# Enforcing Interpretability in Time Series Transformers: A Concept Bottleneck Framework

## Abstract

There has been a recent push of research on Transformer-based models for long-term time series forecasting, even though they are inherently difficult to interpret and explain. While there is a large body of work on interpretability methods for various domains and architectures, the interpretability of Transformer-based forecasting models remains largely unexplored. To address this gap, we develop a framework based on Concept Bottleneck Models to enforce interpretability of time series Transformers. We modify the training objective to encourage a model to develop representations similar to predefined interpretable concepts. In our experiments, we enforce similarity using Centered Kernel Alignment, and the predefined concepts include time features and an interpretable, autoregressive surrogate model (AR). We apply the framework to the Autoformer model, and present an in-depth analysis for a variety of benchmark tasks. We find that the model performance remains mostly unaffected, while the model shows much improved interpretability. Additionally, interpretable concepts become local, which makes the trained model easily intervenable. As a proof of concept, we demonstrate a successful intervention in the scenario of a time shift in the data, which eliminates the need to retrain.

## 1 Introduction

Transformers have shown great success for various types of sequential data, including the modalities of language (Devlin (2018), Brown (2020)), images (Dosovitskiy et al. (2021), Liu et al. (2021)), and speech (Baevski et al. (2020), Gulati et al. (2020)). Their ability to capture long-term dependencies has triggered much interest in applying them to time-series forecasting, for which sequential data is central, and in particular to the challenging task of long-term time series forecasting. Transformer-based architectures, indeed, often show superior performance in this domain (Zhou et al., 2021; 2022; Wu et al., 2021; Ni et al., 2023; Chen et al., 2024), for an overview see Wen et al. (2023).

However, due to their deep and complex architecture, Transformers are difficult to interpret, which is especially important in high-stakes domains such as finance and energy demand prediction. There is a large body of work in the field of Explainable AI to make neural networks more interpretable, including the approach of Concept Bottleneck Models (CBMs) (Koh et al., 2020). This approach relies on the idea of constraining the model such that it predicts human-interpretable concepts first (i.e., the concept bottleneck), and then uses only these concepts to make the final prediction. In order to operationalise this idea, however, concept annotations are needed during training. To overcome this limitation, various approaches have been proposed to learn the concepts themselves using multimodal models (Yuksekgonul et al., 2023; Oikarinen et al., 2023). CBMs and their variants have become popular in different domains, especially computer vision. Nonetheless, as of yet, their application to the time series domain is left underexplored.

In this paper, we propose a domain-agnostic training framework to make any time series Transformer into a Concept Bottleneck Model using time-series specific, yet domain-agnostic concepts, as shown in Figure 1. The first concept is a simple linear surrogate model which is trained on forecasting the same data. Due to its linear nature, the model's predictions are easier to interpret. The second concept is timestamp information provided with the time series data. A key aspect of our training

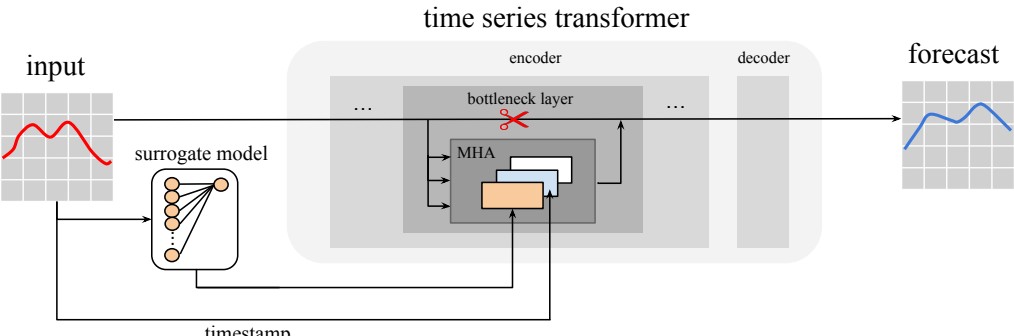

Figure 1: Overview of the concept bottleneck framework. We use one encoder layer as bottleneck, and train its similarity with pre-defined, interpretable concepts. In this example, the bottleneck is located in the multi-headed attention (MHA) block, of which one head is trained to be similar to the surrogate model, another head to the timestamps, and the final head remains untouched.

framework is to leave the model's architecture intact, while encouraging the learned representations to be similar - but not identical - to the interpretable concepts. We use Centered Kernel Alignment to measure the similarity of the bottleneck components with the interpretable concepts, and include it in the training objective. This aspect distinguishes our set-up from the CBM by Koh et al. (2020), which solves the down-stream task by applying a regressor or classifier to the interpretable concepts.

We apply the concept bottleneck framework to the Autoformer model (Wu et al., 2021), which, being common and influential among the time series Transformers, serves as a good representative. We train on six different benchmark datasets, and show that the overall performance remains largely unaffected – in many cases surpassing results from the original Autoformer paper. We present an in-depth interpretability analysis of a trained model, including an analysis of its CKA scores with different concepts and visualizations of the contributions from different components. Finally, we present an intervention in the scenario of a time shift in the data by editing the representations in the bottleneck. Our contributions are summarized as follows:

1. We present a training framework based on Concept Bottleneck Models to enforce interpretability of a time series Transformer.
2. We evaluate the presented framework on the Autoformer model and provide an interpretability analysis.
3. We show how our framework can be used to locally intervene in the model in the scenario of a temporal data shift in the interpretable concepts.

## 2 BACKGROUND AND RELATED WORK

This paper combines and builds upon foundational works from different domains, including Concept Bottleneck Models (CBMs), knowledge transfer with Centered Kernel Alignment (CKA) and time series Transformers. CBMs have been applied to the time series domain before (Ferfoglia et al., 2024), but not with the same interpretable concepts. Likewise, the similarity index CKA has been used before to transfer knowledge between models (Tian et al., 2023), yet, to the best of our knowledge, it has not been used to construct a CBM. Our work aims to bridge this gap, and thereby provides some unique contributions, such as the use of a surrogate model as an interpretable concept.

### 2.1 CONCEPT BOTTLENECK MODELS

Concept Bottleneck Models (CBM; Koh et al., 2020) have emerged in the domain of computer vision as promising interpretable models (Poeta et al., 2023). The concept bottlenecks constrain the model to first predict interpretable concepts, and then use only these concepts in the final downstream task. They are shown to be useful in multiple applications, such as model debugging and human

intervention on decisions. The bottleneck allows for explaining which information the model is using and when it makes an error due to incorrect concept predictions.

However, in standard CBMs, concept annotations are needed during training to learn the bottleneck. To release this restriction, variants have been proposed to learn the concept bottleneck itself too. Yuksekgonul et al. (2023) obtain concept annotations from other datasets or from natural language descriptions of concepts via multimodal models, and Oikarinen et al. (2023) and Yang et al. (2023) first obtain the concept set using GPT-3 and then use a multimodal model to obtain scores for different combinations of input images and concepts. Shang et al. propose incremental concept discovery, such that missing concepts to any concept bank can be identified. However, concept labels do not necessarily contain all information needed to accurately perform the downstream task, and can therefore decrease the task accuracy (Mahinpei et al., 2021). Therefore, Zarlenga et al. (2022) propose Concept Embedding Models, where concepts are represented as a supervised vector, such that richer and more meaningful concept semantics can be captured. Barbiero et al. (2023) propose the Deep Concept Reasoner, which builds symbolic rule structures, using similar high-dimensional concept embeddings, such that the decision process based on the high-dimensional concepts is also interpretable.

CBMs can suffer from information leakage, where the model makes use of additional information in the concept space rather than the concept information itself (Mahinpei et al. (2021), Havasi et al. (2022)). This can occur when the model is jointly trained on the concept prediction and downstream task, and if it uses soft concepts (numerical representations with values between 0 and 1). Information leakage compromises the interpretability and intervenability in soft CBMs, but it can be addressed when introducing a side-channel (Havasi et al., 2022) or by alignment of the model's representation with an underlying data generation process using disentangled representation learning (Marconato et al., 2022).

CBMs and their variants are usually applied to the domain of computer vision, and less frequently to the domain of natural language (Tan et al., 2024), graphs (Barbiero et al., 2023) or tabular data (Zarlenga et al., 2022). In principle, the methodology can be applied to the domain of time series as well, but defining high-level, meaningful concepts is challenging. Ferfoglia et al. (2024) use Signal Temporal Logic (STL) formulas as concept embeddings for time series of cyber-physical systems (recorded by sensors). STL is a formalism to describe the characteristics of time series data and contains the temporal operators 'eventually', 'globally' and 'until', besides the classical propositional logic operators. STL formulas can be converted into a natural language description such as "the temperature should never exceed *a certain threshold* for more than *a specified duration*", and can therefore be human-interpretable concepts. The authors use these concepts as bottleneck for anomaly detection in time series.

## 2.2 Knowledge Transfer with Centered Kernel Alignment

Inspired by neuroscience, Centered Kernel Alignment (CKA) measures the similarity between different representations from neural networks (Kornblith et al., 2019). CKA captures intuitive notions of similarity between representations. It can be used to identify (1) correspondences between layers in the same network trained from different initializations, (2) correspondences in layers across different architectures and (3) across models trained on different datasets. To obtain the score, firstly, the similarity between every pair of examples in each representation separately is measured using a pre-defined kernel, and then the obtained similarity structures are compared. We refer to Kornblith et al. (2019) for more details.

The CKA score can be used to transfer knowledge between different models when included in the loss function, which is shown by Tian et al. (2023). In this work, the authors study Knowledge Distillation between a teacher and student model, and incorporate CKA into the loss function to transfer feature representation knowledge from the pretrained model (teacher) to the incremental learning model (student). The goal is to encourage the incremental learning model not to forget previously learned knowledge, while continuously learning new knowledge, so that it is able to adapt to a dynamic environment (Parisi et al., 2019).

## 2.3 AUTOFORMER

The Autoformer is a Transformer-based model for long-term time series forecasting, as introduced by Wu et al. (2021). The model's encoder-decoder architecture is inspired from time series decomposition, and contains two major modifications to the original Transformer architecture. Firstly, the Autoformer contains decomposition blocks, such that the long-term trend information can be separated from the seasonal information. Secondly, Autoformer employs an auto-correlation mechanism instead of self-attention, such that similarities can be measured on subseries [1]. An overview of the architecture from the original paper is provided in Appendix A.

More specifically, the Autoformer can be regarded as a function $f : \mathbb{R}^{I \times d} \times \mathbb{R}^{I \times 4} \times \mathbb{R}^{O \times 4} \to \mathbb{R}^{O \times d}$, where $I$ is the number of input time steps, $O$ is the number of future time steps, and $d$ is the number of variables in the time series. The first and the second component of the input correspond to *values* denoted as $\boldsymbol{X} \in \mathbb{R}^{I \times d}$, and *timestamps* denoted as $\boldsymbol{T} \in \mathbb{R}^{I \times 4}$, respectively. The additional four dimensions of the latter represent four time features at each time step which are *hour-of-day, day-of-week, day-of-month*, and *day-of-year*. The third component of the model input corresponds to the future time stamps, for which the output values will be forecasted. Note that this set-up is identical to the original paper, however, we explicitly introduce a notation for the timestamps to later define the CKA scores and the intervention.

The Autoformer consists of an encoder and a decoder, which are both constructed from one or multiple layers. The input $\boldsymbol{X}_{en}^0$ for the encoder is defined as:

$$\boldsymbol{X}_{en}^0 = \text{Embedding}(\boldsymbol{X}, \boldsymbol{T}),$$

where $\text{Embedding}(\boldsymbol{X}, \boldsymbol{T}) = \text{Embedding}(\boldsymbol{X}) + \text{Embedding}(\boldsymbol{T})$ where "+" is the entrywise addition after embedding to the same space. The encoder focuses on modelling the seasonality $\boldsymbol{S} \in \mathbb{R}^{I \times d}$, therefore, the trend-cyclical output from the decomposition blocks is eliminated in the encoder, and used only in the decoder. The output $\boldsymbol{X}_{en}^\ell$ of any encoder layer $\ell$ is defined as follows:

$$\boldsymbol{X}_{en}^\ell = \text{Encoder}(\boldsymbol{X}_{en}^{\ell-1}) = \boldsymbol{S}_{en}^{\ell,2},$$
$$\boldsymbol{S}_{en}^{\ell,2}, \_ = \text{SeriesDecomp}(\text{FeedForward}(\boldsymbol{S}_{en}^{\ell,1}) + \boldsymbol{S}_{en}^{\ell,1}),$$
$$\boldsymbol{S}_{en}^{\ell,1}, \_ = \text{SeriesDecomp}(\text{Auto-Correlation}(\boldsymbol{X}_{en}^{\ell-1}) + \boldsymbol{X}_{en}^{\ell-1}),$$

where "_" is the eliminated trend part, and

$$\text{SeriesDecomp}(\boldsymbol{X}) = (\boldsymbol{X}_{seas}, \boldsymbol{X}_{trend}),$$
$$= (\boldsymbol{X} - \text{AvgPool}(\text{Padding}(\boldsymbol{X})), \text{AvgPool}(\text{Padding}(\boldsymbol{X}))),$$
$$\text{FeedForward}(\boldsymbol{S}_{en}^\ell) = \max(0, \boldsymbol{S}_{en}^\ell \boldsymbol{W}_1^\ell + \mathbf{b}_1^\ell)\boldsymbol{W}_2^\ell + \mathbf{b}_2^\ell,$$
$$\text{Auto-Correlation}(\boldsymbol{X}_{en}^\ell) = \boldsymbol{W}_0^{\ell+1} \cdot \text{Concat}\left(\text{head}_1^{\ell+1}(\boldsymbol{X}_{en}^\ell), \ldots, \text{head}_h^{\ell+1}(\boldsymbol{X}_{en}^\ell)\right).$$

Note that the AvgPool function finds the moving average, and the Padding function keeps the series length unchanged. Also, note that the FeedForward function consists of two linear functions with a ReLU activation function in between. For future reference, we denote its output as follows: $\text{FeedForward}(\boldsymbol{S}_{en}^\ell) = \boldsymbol{Z}^\ell \in \mathbb{R}^{d_1 \times d_2}$. The Auto-Correlation function consists of $h$ auto-correlation heads. The matrix $\boldsymbol{W}_0$ is multiplied to their concatenated outputs for projection. We omit the definition of the decoder, because our bottleneck framework does not include it. For details on the decoder and the implementation, we refer the reader to Wu et al. (2021).

## 3 METHOD

We propose a framework to make the Autoformer model interpretable by including a bottleneck based on knowledge transfer with CKA (Kornblith et al., 2019), as shown in Figure 2. The basic idea is that we assign one of the encoder layers to be the *bottleneck*, of which we calculate the CKA scores with the interpretable concepts. These CKA scores are included in the loss function to encourage the model to learn the interpretable concepts.

---

[1]Note that the *auto-correlation block* is a specific implementation of the *attention block*, so we use the two terms interchangeably.

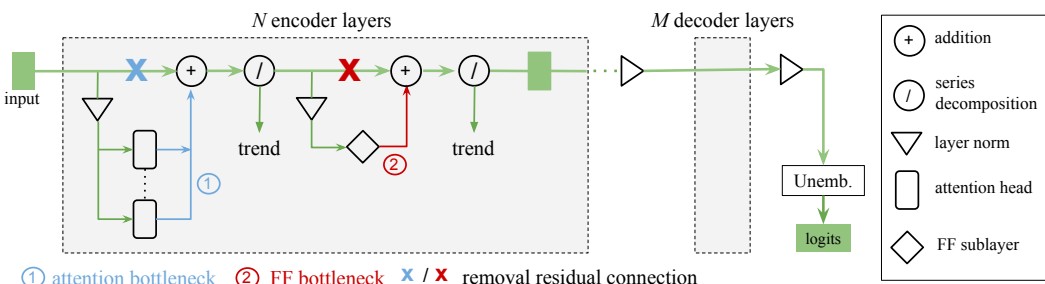

Figure 2: Architecture of Autoformer with a concept bottleneck in the attention mechanism (blue) or the FF network (red). Note that the residual connection is removed at the location of the bottleneck (and the 'residual stream' thus interrupted). The series decomposition blocks are characteristic for the Autoformer. (Visualisation inspired by Rai et al., 2024).

## 3.1 LOSS FUNCTION

The loss function should encourage the model to represent the interpretable concepts in the bottleneck layer. Therefore, we add a term $\mathcal{L}_{CKA}$ to the loss function based on the CKA scores of the bottleneck and the interpretable concepts. In particular, low similarity between the bottleneck and the interpretable concepts results in a higher value for $\mathcal{L}_{CKA}$. The total loss function $\mathcal{L}_{Total}$, then, is a weighted average of the Mean Squared Error (MSE) loss $\mathcal{L}_{MSE}$ and the CKA loss $\mathcal{L}_{CKA}$:

$$\mathcal{L}_{Total} = (1 - \alpha)\, \mathcal{L}_{MSE} + \alpha\, \mathcal{L}_{CKA}, \tag{1}$$

$$\mathcal{L}_{CKA} = 1 - \frac{1}{c} \sum_{i=1}^{c} CKA_i, \tag{2}$$

with $\alpha$ as a hyperparameter, $c$ denoting the number of concepts, and $CKA_i \in [0, 1]$ denoting the CKA score between the model's representation and concept $i$. More details on the exact calculation of the CKA score are described in Section 3.2.

## 3.2 INTERPRETABLE CONCEPTS IN THE BOTTLENECK

To measure the presence of a concept in the bottleneck, we need to introduce a score based on CKA. In this section, we first describe the location of the representations in the bottleneck layer. Next, we describe which interpretable concepts are used, and how all the information is enforced to pass through the bottleneck components.

**Location bottleneck.** We assign one encoder layer to be the bottleneck layer, because the encoder focuses on modelling seasonal information from the data. Within the bottleneck layer, the Autoformer's representations can be taken from two different types of blocks: the auto-correlation block and the feed-forward block. Therefore, we refer to two types $\tau$ of bottlenecks corresponding to where the latent representations are received from: the attention bottleneck ($\tau = Att$) and the feed-forward bottleneck ($\tau = FF$), respectively.

Let the bottleneck layer $\hat{\ell}$, of type $\tau$, contain $c$ latent representations or *components*, i.e., $\left(\boldsymbol{H}_i^{\hat{\ell}}\right)_{i=1}^{c}$.
Note that these are not the same as the output of the bottleneck layer. Each component $\boldsymbol{H}_i^{\hat{\ell}}$ should represent a pre-defined interpretable concept. Since the attention block is multi-headed, different heads naturally form the components of the attention bottleneck. Moreover, the components need to be divided between the heads, which would be convenient when the number of heads is a multiple of the total number of concepts to maintain a uniform concept per head ratio. For the feed-forward bottleneck, we define the components to be slices from its output $\boldsymbol{Z}^{\hat{\ell}} \in \mathbb{R}^{d_1 \times d_2}$, such that stacking the components results in the original output. In other words, each concept is imposed on a single, contiguous latent representation slice $\boldsymbol{Z}_i^{\hat{\ell}}$, where we slice in the first dimension $d_1$. Using the

previously defined notation in Section 2.3, a complete expression for the component $\boldsymbol{H}_{\tau,i}^{\hat{\ell}}$ is:

$$\boldsymbol{H}_{\tau,i}^{\hat{\ell}} = \begin{cases} \text{head}_i^{\hat{\ell}}(\boldsymbol{X}_{en}^{\hat{\ell}-1}) & \text{if bottleneck type } \tau = Att, \\ \boldsymbol{Z}_i^{\hat{\ell}} & \text{if bottleneck type } \tau = FF. \end{cases}$$

We refer to Appendix C for detailed visualizations of both types of bottlenecks.

**Interpretable concepts.** We use two domain-agnostic interpretable concepts which can be used for forecasting, namely: (1) a simple, human-interpretable surrogate forecasting model, (2) the input timestamps recorded with the time series.

1. We use a simple autoregressive model (AR) as a surrogate model, which predicts the next future value as a linear combination of its past values. This model is transparent, and the attribution of each input feature to the output can be simply interpreted by its weight. This concept can also be regarded as a baseline for the forecasting performance. The model is fit to the same training data as the Autoformer model.
2. We use the hour-of-day feature from the timestamps $\boldsymbol{T}$ as interpretable time concept, denoted by $\boldsymbol{T}_{hourofday}$. This provides the bottleneck with a simplified notion of time.

The CKA scores for these concepts can thus be calculated as follows, respectively:

$$CKA_{AR}(\boldsymbol{X}, \boldsymbol{T}, \tau) = CKA\left(\boldsymbol{H}_{\tau,i=1}^{\hat{\ell}}, \text{AR}(\boldsymbol{X})\right),$$

$$CKA_{time}(\boldsymbol{X}, \boldsymbol{T}, \tau) = CKA\left(\boldsymbol{H}_{\tau,i=2}^{\hat{\ell}}, \boldsymbol{T}_{hourofday}\right).$$

**Removal of residual connection.** Any Autoformer layer contains residual connections around the auto-correlation and feed-forward blocks. To ensure that all information passes through the bottleneck, we remove the residual connection around the bottleneck, potentially at the cost of a loss in performance. Otherwise, any concept, including the interpretable concepts, can be passed through the residual connection and compromise the effectiveness of the bottleneck. This modifies either the auto-correlation or the feed-forward block in bottleneck $\hat{\ell}$, depending on the type, as follows:

$$\boldsymbol{S}_{en}^{\hat{\ell},2}, \_ = \text{SeriesDecomp}(\text{FeedForward}(\boldsymbol{S}_{en}^{\hat{\ell},1})), \qquad \text{if bottleneck type } \tau = FF,$$

$$\boldsymbol{S}_{en}^{\hat{\ell},1}, \_ = \text{SeriesDecomp}(\text{Auto-Correlation}(\boldsymbol{X}_{en}^{\hat{\ell}-1})), \qquad \text{if bottleneck type } \tau = Att.$$

In the scenario that the number of components is equal to the number of interpretable concepts ($c = 2$), the construction of the bottleneck limits learning domain-specific features from the data, other than the interpretable concepts. Therefore, we perform experiments where we allow an extra component in the bottleneck to not learn any pre-defined concept ($c = 3$). In other words, the extra component serves as a *side-channel* or *free component*. No CKA loss is calculated using this component, and therefore this training set-up can be regarded as semi-supervised. The free component may partly restore what we lost by removing the residual connection, but with the advantage that we can monitor which information goes through it, and even visualize it (as in Section 4.2.2).

**Implementation details.** In our experiments, we use an Autoformer with three encoder layers, of which the bottleneck layer is at position $\ell = 2$. Similar to the original Autoformer paper, we use one decoder layer, employ the Adam optimizer (Kingma & Ba, 2017) with an initial learning rate of $10^{-4}$, and use a batch size of 32. The training process is early stopped within 25 epochs. All experiments are repeated five times on different seeds, using hyperparameter $\alpha = 0.3$.

# 4 EXPERIMENTS

We use our framework to evaluate the Autoformer model on six real-world benchmarks in the domains of energy, traffic, economics, weather, and disease, similar to Wu et al. (2021). For more information on the datasets, we refer the reader to Appendix B. We train the model with and without the bottleneck, using different configurations for the bottleneck. Before training the Autoformer with a bottleneck, we train a simple AR model on the same data, so that its outputs can be used to align the representations of the bottleneck. Additionally, to demonstrate the generality of the framework, we repeat the experiments for the vanilla Transformer architecture in Appendix H and on a synthetic dataset in Appendix I.

## 4.1 PERFORMANCE ANALYSIS

The performance of different models (on held-out test sets) is shown in Table 1. We compare Autoformers trained with our bottleneck framework (containing a free component, i.e., $c = 3$) against the AR surrogate model and the Autoformer model as mentioned in the original paper from Wu et al. (2021), which contains two encoder layers and eight heads per layer. Visualizations of the forecasts from these models are shown in Appendix D.

Surprisingly, the surrogate AR model outperforms the other models for most datasets w.r.t. both the Mean Squared Error (MSE) and the Mean Absolute Error (MAE), suggesting that there are many linear relationships in these time series. Even though this model is very simple, it is able to forecast most time series with lower error scores.[2] Additionally, our Autoformer without bottleneck (labeled **No bottleneck**) performs better than the one from Wu et al. (labeled **Original**), which may be explained by the slightly increased model size. Furthermore, models with bottlenecks typically show slightly worse performance, with the difference varying by dataset. More detailed results are presented in Appendix E and F, where the first includes the results for bottlenecks without free component (including the standard deviation for different seeds), and the latter includes a sensitivity analysis to hyperparameter $\alpha$.

Table 1: Performance of different models. For both metrics, it holds that a lower score indicates a better performance, where the best results are **bold**, and the second-best are underlined.

| | Att bottleneck | | FF bottleneck | | No bottleneck | | AR | | Original | |
|---|---|---|---|---|---|---|---|---|---|---|
| | MSE | MAE | MSE | MAE | MSE | MAE | MSE | MAE | MSE | MAE |
| Electricity | 0.231 | 0.338 | 0.207 | 0.320 | 0.280 | 0.368 | 0.497 | 0.522 | **0.201** | **0.317** |
| Traffic | 0.642 | 0.393 | **0.393** | **0.377** | 0.619 | 0.387 | 0.420 | 0.494 | 0.613 | 0.388 |
| Weather | 0.290 | 0.354 | 0.271 | 0.341 | 0.269 | 0.344 | **0.006** | **0.062** | 0.266 | 0.336 |
| Illness | 3.586 | 1.313 | 3.661 | 1.322 | 3.405 | 1.295 | **1.027** | **0.820** | 3.483 | 1.287 |
| Exchange rate | 0.195 | 0.323 | 0.155 | 0.290 | 0.152 | 0.283 | **0.082** | **0.230** | 0.197 | 0.323 |
| ETT | 0.177 | 0.282 | 0.174 | 0.280 | 0.155 | 0.265 | **0.034** | **0.117** | 0.255 | 0.339 |

## 4.2 INTERPRETABILITY ANALYSIS

To demonstrate the impact of the bottleneck in the Autoformer model, we first conduct a CKA analysis on the bottleneck layer with the corresponding interpretable concepts, and then visually demonstrate how each component contributes to the final forecast.

### 4.2.1 CKA ANALYSIS

To test the extent to which the bottleneck represents the interpretable concepts, we calculate the CKA scores of the model's representations with the concept representations. The scores of the attention and feed-forward bottleneck on the electricity dataset are shown in Figure 3 (see Figure 25d in Appendix F for the scores without bottleneck). Both models contain three heads per layer, with `layer1` as bottleneck, of which the bottom, middle and upper layers in the figure correspond to the AR, hour-of-day, and free component, respectively. For instance, we obtain a similarity of 0.58 between the first component of the attention bottleneck and AR, which is moderately high (recall that CKA scores range from 0 for totally dissimilar to 1.0 for identical, although potentially scaled and rotated).

The scores in the bottleneck indicate that the representations are not completely similar to the intended concepts, e.g., the attention bottleneck has a similarity of 0.58 with the AR model, and 0.99 with the hour-of-day feature. The similarity of the bottleneck's representations with the intended concepts also depends on the bottleneck's location, as the feed-forward bottleneck shows a similarity of 0.61 with AR and a similarity of 0.77 with hour-of-day. These results indicate that the training

---

[2]Note that the phenomenon that simple models sometimes beat time series Transformers (Zeng et al., 2022) has been observed before. There has been a vivid discussion about the relevance of these results, for instance here. These discussions are beyond the scope of our paper, which rather targets interpretability of time series Transformers. For more information on the effect of AR as surrogate model, see Appendix J.

framework can encourage the components to form representations that are perfectly similar to the interpretable concepts, even though this is not always realized for each component.

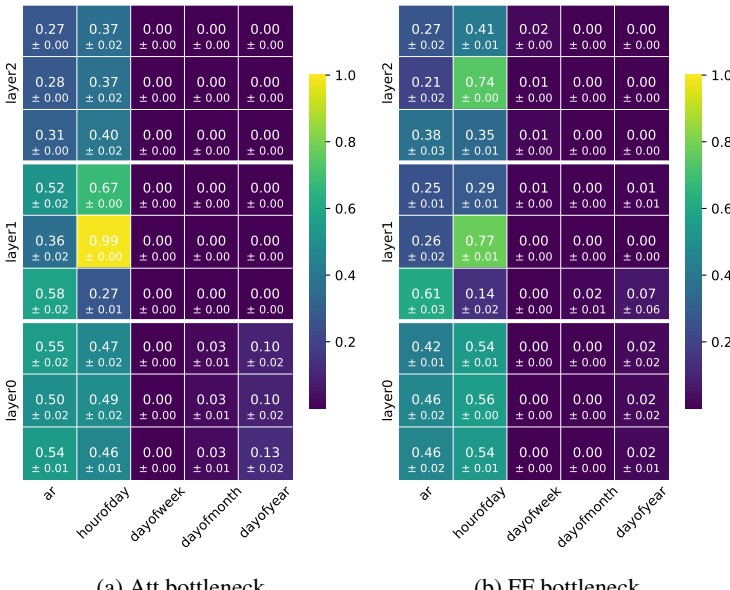

(a) Att bottleneck          (b) FF bottleneck

Figure 3: CKA scores of the encoder (containing three heads per layer) from the attention and feed-forward bottleneck on the electricity dataset, where each score denotes the similarity of an individual component. The first component of `layer1` is trained to be similar to AR, and the second component to the 'hour-of-day' concept (lower and middle row in the figure, respectively). The scores are calculated using three batches of size 32 from the test data set.

### 4.2.2 COMPONENT VISUALIZATIONS

Interestingly, because the components we consider all read and write from the 'residual stream' (Elhage et al., 2021), we can visualize the contributions to the final prediction of each component separately by applying the entire Transformer-Decoder to the component representations. This is the Decoder Lens method, described in detail in Langedijk et al. (2023). Using this method, we obtain visualizations of the contributions of each component in the bottleneck, see Figure 4. We obtain the output from the full bottleneck by applying the decoder to the output of the bottleneck (after performing layer normalization). The output from each component individually is obtained by masking the other components with zero (close to the mean).

From Figure 4a and 4b we see that the different bottleneck components are similar to the concepts they were trained on. In particular, the first component shows a forecast with correct periodicity and few irregularities, similar to the actual forecast from the AR model. Likewise, the second component shows a periodicity to the actual hour-of-day feature. The third component is not trained to be similar to an interpretable concept, and seems to pick up on the high-frequency patterns in the data, e.g., the low, second peak in the forecast. This observation is further strengthened by Figure 20f, which shows that the final forecast consists of many high-frequency patterns when using only the third component from the bottleneck.

### 4.3 INTERVENTION

One benefit of interpreting trained models is gaining a deeper understanding and, possibly, more control of the model's behavior. This can be useful in the scenario of out-of-distribution data at inference time. If the data changes in features that can be interpreted in the model, it is feasible to intervene locally in these concepts to exclusively employ the model with data from its training distribution. To show such benefit of our framework, we evaluate the trained model on data with shifted timestamps and compare it with performing an intervention on the shifted concept.

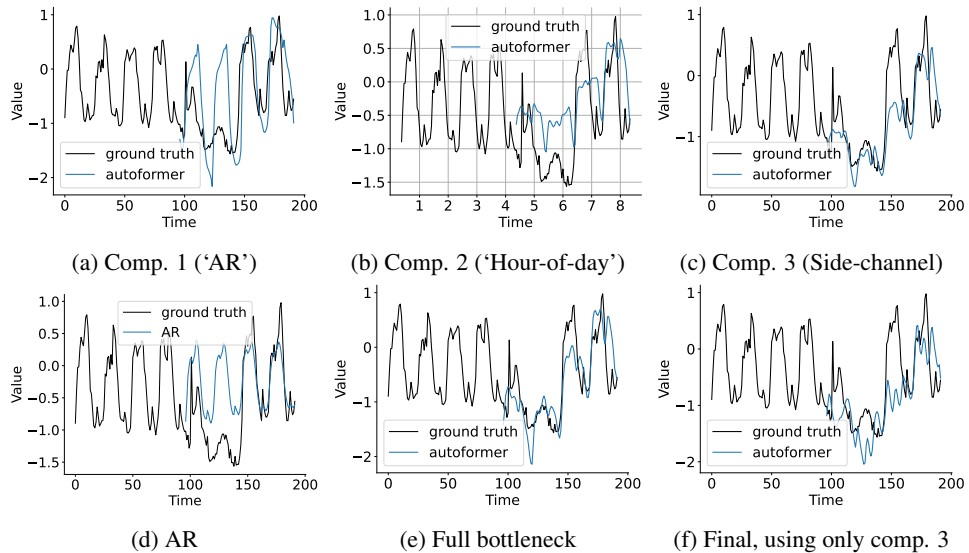

(a) Comp. 1 ('AR')  (b) Comp. 2 ('Hour-of-day')  (c) Comp. 3 (Side-channel)

(d) AR  (e) Full bottleneck  (f) Final, using only comp. 3

Figure 4: Forecasts from the components in the bottleneck layer (FF bottleneck on electricity data) in 4a, 4b and 4c. They are obtained by masking the other components with zero (the mean). The first half of the ground truth forms the input to the model. Note that the horizontal axes are the same across all figures, but Figure 4b contains a grid of days instead of numbered hours. Figure 20d shows the forecast made by the surrogate model AR; Figure 20e shows the forecast of the entire layer (i.e., all components together), and 20f shows the forecast of the final layer when only the third component is used in the bottleneck layer. Note the difference between Figures 4c and 20f, where we decode from the bottleneck and the final layer, respectively. All forecasts are obtained using the Decoder Lens method Langedijk et al. (2023).

We delay the input timestamps $\boldsymbol{T} \in \mathbb{R}^{I \times 4}$ with a fixed number of hours to obtain the shifted timestamps $\widetilde{\boldsymbol{T}}$, so that the patterns associated to the hour-of-day feature are misleading. We run the model on both types of timestamps, and perform an intervention in the bottleneck by substituting the activations based on the shifted time with the activations based on the original time.

We define the intervention by introducing $\widetilde{\boldsymbol{X}}_{en}^{\ell}$ as the output of any encoder layer $\ell$ with input $\widetilde{\boldsymbol{T}}$:

$$\widetilde{\boldsymbol{X}}_{en}^{\ell} = \begin{cases} \text{Embedding}(\boldsymbol{X}, \widetilde{\boldsymbol{T}}) & \text{if } \ell = 0, \\ \text{Encoder}(\widetilde{\boldsymbol{X}}_{en}^{\ell-1}) = \widetilde{\boldsymbol{S}}_{en}^{\ell,2} & \text{otherwise.} \end{cases}$$

The definition of the output $\boldsymbol{X}_{en}^{\ell}$ of any encoder layer $\ell$ remains unchanged, and is based on $\boldsymbol{T}$.

The key aspect of the intervention is to replace the input $\widetilde{\boldsymbol{X}}_{en}^{\hat{\ell}-1}$ of the bottleneck layer $\hat{\ell}$ with $\boldsymbol{X}_{en}^{\hat{\ell}-1}$, but only in the component that represents the time concept. This can be done in the bottleneck only, because, by construction, its location of the concept representations is known. The intervention is minimal, since the input to most model component remains unchanged, except for one component in the bottleneck layer. If type $\tau = Att$, we intervene on the attention block in the bottleneck, or if type $\tau = FF$, we intervene on the feed-forward block. That is, the function Auto-Correlation$_{\text{Int}}$ for a bottleneck of type $\tau = Att$ is defined as follows:

Auto-Correlation$_{\text{Int}}(\boldsymbol{X}_{en}^{\ell}, \widetilde{\boldsymbol{X}}_{en}^{\ell}) =$

$$\begin{cases} \boldsymbol{W}_0^{\ell+1} \cdot \text{Concat}\left(\text{head}_1^{\ell+1}(\widetilde{\boldsymbol{X}}_{en}^{\ell}), \text{head}_2^{\ell+1}(\boldsymbol{X}_{en}^{\ell}), \text{head}_3^{\ell+1}(\widetilde{\boldsymbol{X}}_{en}^{\ell})\right) & \text{if } \ell = \hat{\ell}, \\ \text{Auto-Correlation}(\boldsymbol{X}_{en}^{\ell}) & \text{otherwise,} \end{cases}$$

and the FeedForward$_{\text{Int}}$ function for a bottleneck of type $\tau = FF$ is defined as follows:

$$\text{FeedForward}_{\text{Int}}(\boldsymbol{S}_{en}^{\ell}, \widetilde{\boldsymbol{S}}_{en}^{\ell}) = \begin{cases} \text{Stack}(\widetilde{\boldsymbol{Z}}_1^{\ell}, \boldsymbol{Z}_2^{\ell}, \widetilde{\boldsymbol{Z}}_3^{\ell}) & \text{if } \ell = \hat{\ell}, \\ \text{FeedForward}(\boldsymbol{S}_{en}^{\ell}) & \text{otherwise.} \end{cases}$$

In both functions we make use of the fact that the time concept is represented in the second component, and there are three components in total.

We use a bottleneck Autoformer trained on the electricity dataset, and perform shifts of up to and including 23 hours. We compare the performance of the intervention with out-of-the-box performance of the same model on the shifted dataset. The results are shown in Figure 5.

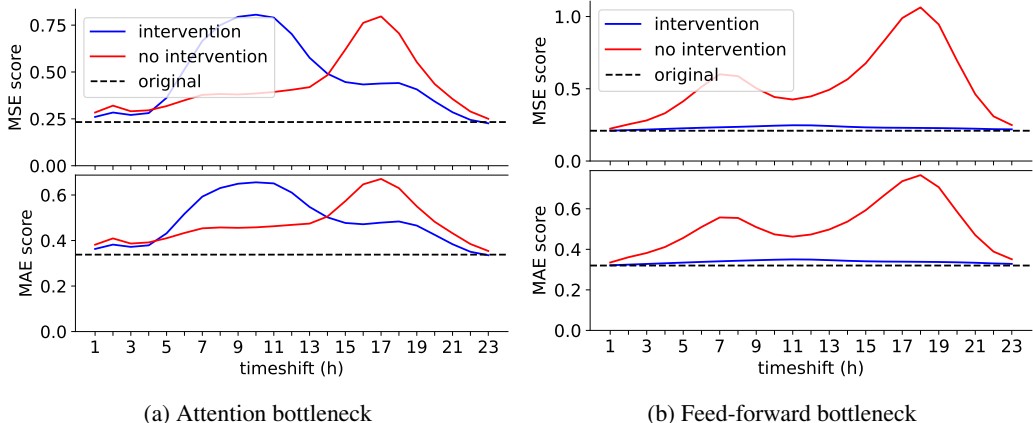

(a) Attention bottleneck    (b) Feed-forward bottleneck

Figure 5: Performance of the bottleneck Autoformer on electricity data with shifted timestamps. The intervention consists of replacing the activations in the bottleneck with activations from the original timestamps (only in the time component). The dashed line represents the performance of the same model on the original data, i.e., with no timeshift.

Interestingly, the two types of bottlenecks show different results under the intervention. Where the intervention hurts performance for smaller timeshifts in the attention bottleneck, it improves performance for all time shifts in the feed-forward bottleneck. In fact, the performance of the intervention is even close to the original performance in the feed-forward bottleneck. These results indicate that the choice of bottleneck location is relevant, and suggest that the concept of time might be represented in a more complex manner in the attention heads. Presumably, intervening in an individual head within the multi-headed auto-correlation mechanism provides more unforeseen consequences than intervening in the slice of a linear layer due to the increased complexity. All in all, the results show that the (feed-forward) bottleneck Autoformer can be effectively employed on the shifted data without re-training, given that the shift is known, and it is shifted in the interpretable concepts.

## 5 DISCUSSION AND CONCLUSIONS

In this work, we propose a training framework based on Concept Bottleneck Models to enforce interpretability of time series Transformers. We introduce a new loss function based on the similarity score CKA of the model's representations and interpretable concepts. We apply the framework to the Autoformer model using six different datasets. Our results indicate that the overall performance remains unaffected, while the model's components become more interpretable. Additionally, it becomes possible to perform a local intervention when employing the model after a temporal data shift. The scope of our study has been limited to a single Transformer model, and an interesting direction for future research would be to extend this framework to other Transformer models and modalities. Additionally, future work could focus on optimizing the number and type of interpretable concepts in the bottleneck. We hope our work contributes to a deeper understanding of (time series) Transformers and their behavior in different domains. In particular, recent progress in the field of mechanistic interpretability is based on the observation that the residual stream of the Transformer encourages modular solutions, which enables localized concepts or specialized circuitry to perform a specific task. Instead of relying on post-hoc localization of these concepts, our paper presents a demonstration that we can encourage locality of concepts, without a significant loss in performance.

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

## A   AUTOFORMER ARCHITECTURE

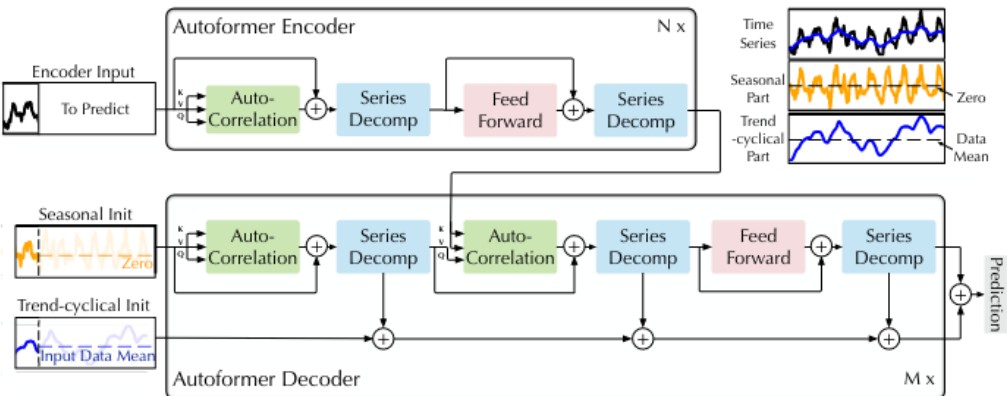

Figure 6: Overview of the Autoformer architecture from Wu et al. (2021). The Autoformer consists of an encoder and decoder, which are built from auto-correlation, series decomposition and feed-forward blocks. The encoder eliminates the long-term trend-cyclical part, while the decoder accumulates the trend part.

## B   DATASETS

We evaluate the Autoformer model on six real-world benchmarks, covering the five domains of energy, traffic, economics, weather, and disease. We use the same datasets as Wu et al. (2021), and provide additional information in Table 2, as given in the original Autoformer paper.

Table 2: Descriptions of the datasets, as given by Wu et al. (2021) and shared online. 'Pred len' denotes the prediction length used in our experiments.

| Dataset | Pred len | Description |
| --- | --- | --- |
| Electricity | 96 | Hourly electricity consumption of 321 customers from 2012 to 2014. |
| Traffic | 96 | Hourly data from California Department of Transportation, which describes the road occupancy rates measured by different sensors on San Francisco Bay area freeways. |
| Weather | 96 | Recorded every 10 minutes for 2020 whole year, which contains 21 meteorological indicators, such as air temperature, humidity, etc. |
| Illness | 24 | Includes the weekly recorded influenza-like illness (ILI) patients data from Centers for Disease Control and Prevention of the United States between 2002 and 2021, which describes the ratio of patients seen with ILI and the total number of the patients. |
| Exchange rate | 96 | Daily exchange rates of eight different countries ranging from 1990 to 2016. |
| ETT | 96 | Data collected from electricity transformers, including load and oil temperature that are recorded every 15 minutes between July 2016 and July 2018. |

## C  DETAILED OVERVIEW BOTTLENECK ARCHITECTURE

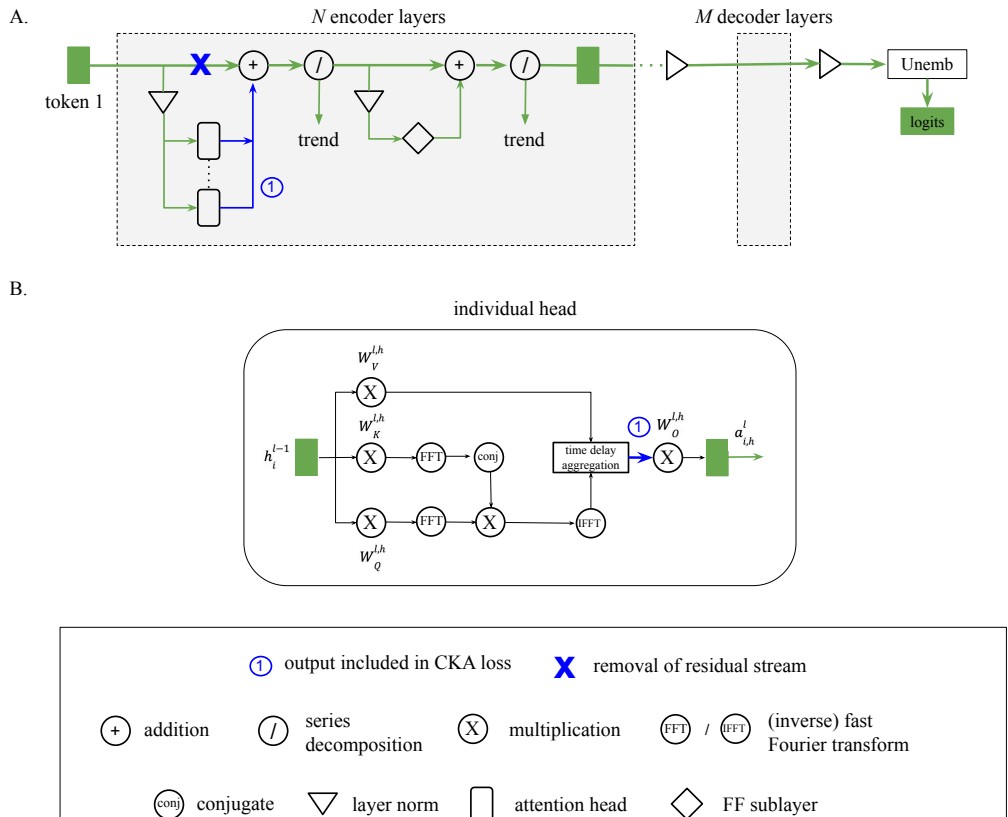

Figure 7: Architecture of Autoformer with a concept bottleneck in the attention mechanism. Figure A shows the total architecture, including the removal of the residual stream at the location of the attention block. Figure B shows the overview of an individual attention head $i$ with its input representation $h_i^{l-1}$ and output activation $a_{i,h}^l$. The representations after the time delay aggregation block are used in the bottleneck, because these correspond to the output of head $i$. Note that in practice, the individual head representations do not exist anymore after multiplication with the weights $W_o^{l,h}$, as this matrix projects the activations of all heads together to the model dimension.

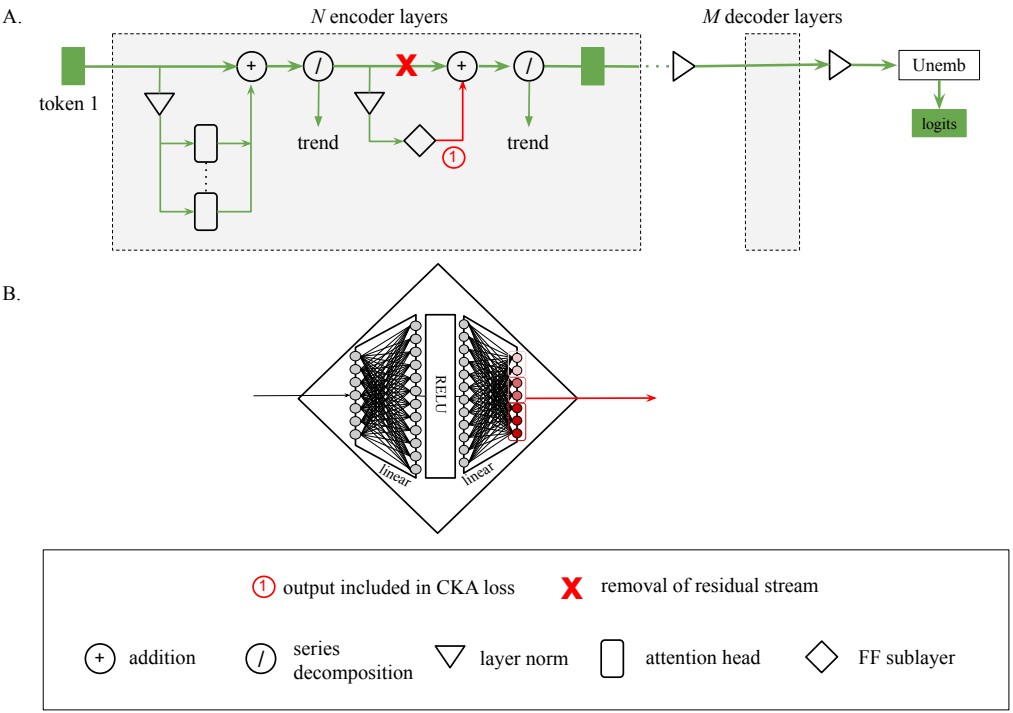

Figure 8: Architecture of Autoformer with a concept bottleneck in the feed-forward sublayer. Figure A shows the total architecture, including the removal of the residual stream at the location of the feed-forward sublayer. Figure B shows the overview of the feed-forward network consisting of two linear layers connected with the RELU activation function in between. The output of the final linear layer is used in the bottleneck. The output is split into three parts to assign the concepts to different components in the bottleneck.

# D  QUALITATIVE RESULTS

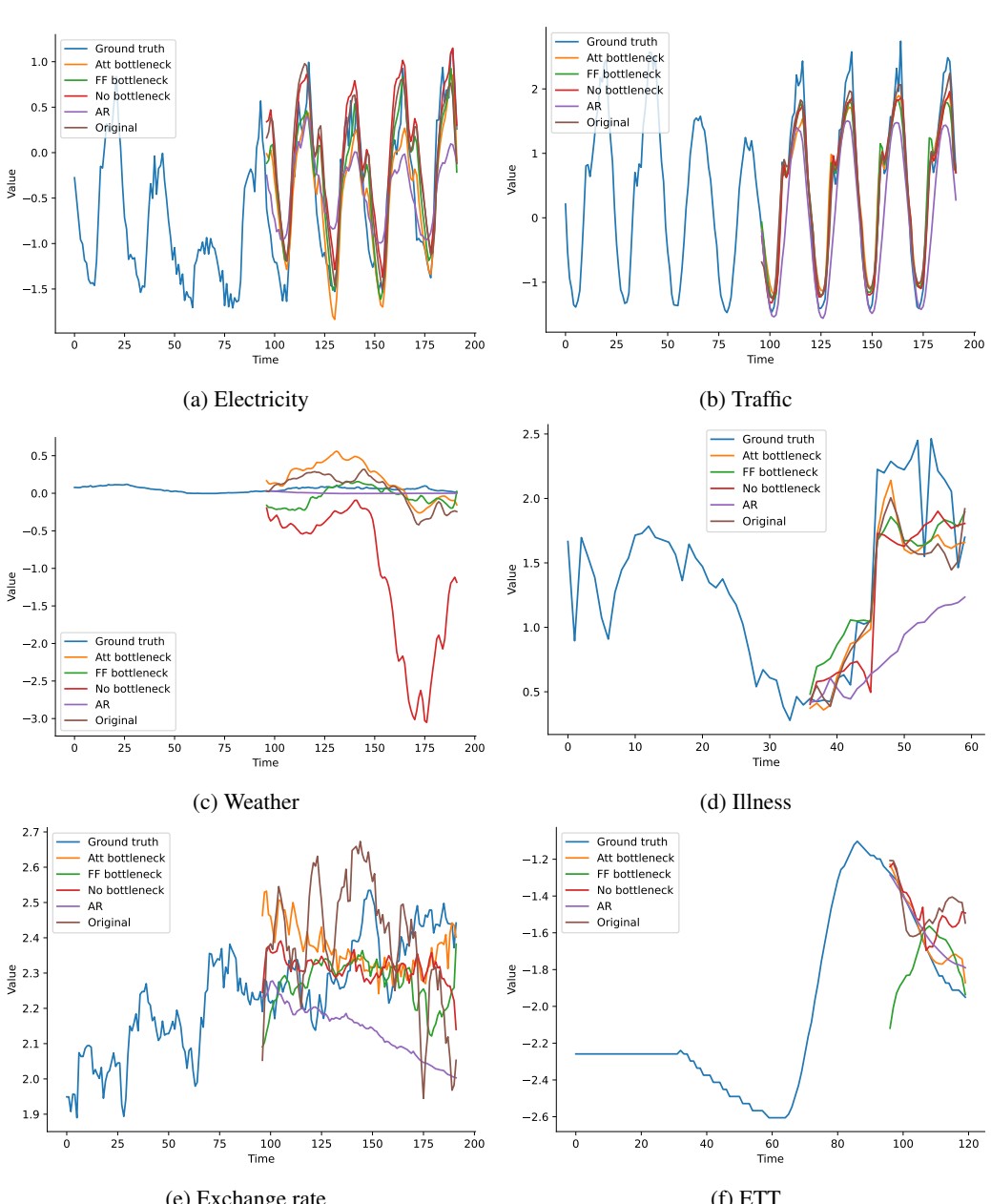

(a) Electricity

(b) Traffic

(c) Weather

(d) Illness

(e) Exchange rate

(f) ETT

Figure 9: Forecasts on different datasets. The first part of the ground truth (shown in blue) is the input for the models, and the test set is used for each dataset.

# E  DETAILED RESULTS

Table 3: Performance of different models in Mean Squared Error (MSE) and Mean Absolute Error (MAE). The bottlenecks do contain a free component ($c = 3$), and use AR as surrogate model. The model with no bottleneck is an original Autoformer of similar size. For all datasets, the shortest prediction lengths from Wu et al. (2021) are used, see Table 2. The standard deviation is determined using five different seeds.

| Free component | Att bottleneck | | FF bottleneck | | No bottleneck | | AR | | Original | |
|---|---|---|---|---|---|---|---|---|---|---|
| | MSE | MAE | MSE | MAE | MSE | MAE | MSE | MAE | MSE | MAE |
| Electricity | 0.231 ± 0.009 | 0.338 ± 0.005 | 0.207 ± 0.005 | 0.320 ± 0.005 | 0.280 ± 0.165 | 0.368 ± 0.111 | 0.497 | 0.522 | **0.201** ± 0.003 | **0.317** ± 0.004 |
| Traffic | 0.642 ± 0.022 | 0.393 ± 0.013 | **0.393** ± 0.013 | **0.377** ± 0.006 | 0.619 ± 0.015 | 0.387 ± 0.005 | 0.420 | 0.494 | 0.613 ± 0.028 | 0.388 ± 0.012 |
| Weather | 0.290 ± 0.027 | 0.354 ± 0.020 | 0.271 ± 0.016 | 0.341 ± 0.011 | 0.269 ± 0.000 | 0.344 ± 0.000 | **0.006** | **0.062** | 0.266 ± 0.007 | 0.336 ± 0.006 |
| Illness | 3.586 ± 0.241 | 1.313 ± 0.040 | 3.661 ± 0.237 | 1.322 ± 0.050 | 3.405 ± 0.208 | 1.295 ± 0.044 | **1.027** | **0.820** | 3.483 ± 0.107 | 1.287 ± 0.018 |
| Exchange rate | 0.195 ± 0.029 | 0.323 ± 0.025 | 0.155 ± 0.010 | 0.290 ± 0.013 | 0.152 ± 0.003 | 0.283 ± 0.003 | **0.082** | **0.230** | 0.197 ± 0.019 | 0.323 ± 0.012 |
| ETT | 0.177 ± 0.003 | 0.282 ± 0.004 | 0.174 ± 0.006 | 0.280 ± 0.005 | 0.155 ± 0.004 | 0.265 ± 0.002 | **0.034** | **0.117** | 0.255 ± 0.020 | 0.339 ± 0.020 |

Table 4: Performance on different datasets, where the bottlenecks do not contain a free component. AR is used as surrogate model in the bottlenecks. The model with no bottleneck is an original Autoformer of similar size. For all datasets, the shortest prediction lengths from Wu et al. (2021) are used, see Table 2. The standard deviation is determined using five different seeds.

| No free component | Att bottleneck | | FF bottleneck | | No bottleneck | | AR | | Original | |
|---|---|---|---|---|---|---|---|---|---|---|
| | MSE | MAE | MSE | MAE | MSE | MAE | MSE | MAE | MSE | MAE |
| Electricity | 0.224 ± 0.006 | 0.332 ± 0.003 | 0.206 ± 0.009 | 0.321 ± 0.009 | 0.202 ± 0.006 | 0.318 ± 0.007 | 0.497 | 0.522 | **0.201** ± 0.003 | **0.317** ± 0.004 |
| Traffic | 0.629 ± 0.023 | 0.394 ± 0.015 | 0.627 ± 0.031 | 0.392 ± 0.025 | 0.613 ± 0.018 | **0.378** ± 0.007 | **0.420** | 0.494 | 0.613 ± 0.028 | 0.388 ± 0.012 |
| Weather | 0.281 ± 0.025 | 0.348 ± 0.018 | 0.260 ± 0.015 | 0.333 ± 0.013 | 0.257 ± 0.004 | 0.332 ± 0.005 | **0.006** | **0.062** | 0.266 ± 0.007 | 0.336 ± 0.006 |
| Illness | 3.966 ± 0.296 | 1.401 ± 0.073 | 3.721 ± 0.268 | 1.351 ± 0.053 | 3.585 ± 0.331 | 1.333 ± 0.070 | **1.027** | **0.820** | 3.483 ± 0.107 | 1.287 ± 0.018 |
| Exchange rate | 0.208 ± 0.026 | 0.333 ± 0.022 | 0.158 ± 0.009 | 0.293 ± 0.009 | 0.152 ± 0.006 | 0.284 ± 0.007 | **0.082** | **0.230** | 0.197 ± 0.019 | 0.323 ± 0.012 |
| ETT | 0.178 ± 0.011 | 0.283 ± 0.007 | 0.174 ± 0.01 | 0.283 ± 0.009 | 0.165 ± 0.004 | 0.274 ± 0.004 | **0.034** | **0.117** | 0.255 ± 0.020 | 0.339 ± 0.020 |

# F   HYPER-PARAMETER SENSITIVITY

To verify the sensitivity to hyperparameter $\alpha$ in the loss function, we train the Autoformer with a feed-forward bottleneck on different values for $\alpha$, where the bottleneck contains a free component ($c = 3$) and the model is trained on the electricity dataset. The results are given in Table 5. Interestingly, a high value for $\alpha$ provides the (slightly) best performance (i.e., $\alpha = 0.7$), which corresponds to a high importance for the bottleneck components to be similar to the interpretable concepts. This verifies that training for interpretability does not necessarily hurt the performance, at least not in this set-up.

Table 5: Performance of the Autoformer for different values of $\alpha$ in MSE and MAE. For both metrics, it holds that a lower score indicates a better performance, where the best results are **bold**, and the second-best are underlined.

| | **FF bottleneck** | |
|---|---|---|
| $\alpha$ | MSE | MAE |
| 0.0 | 0.199 ± 0.004 | 0.315 ± 0.005 |
| 0.3 | 0.207 ± 0.005 | 0.320 ± 0.005 |
| 0.5 | 0.200 ± 0.006 | 0.314 ± 0.006 |
| 0.7 | **0.199** ± 0.002 | **0.313** ± 0.002 |
| 1.0 | 1.177 ± 0.027 | 0.876 ± 0.010 |

Interestingly, the best and second-best values for $\alpha$ are 0.7 and 0.0, respectively. These values are not close, but it should be noted that the error scores for all $\alpha < 1$ are close in value, especially considering the error margins. In other words, the labels of *best* and *second-best* do not carry much of weight. To visualize this, a plot of the same results is given in Figure 10.

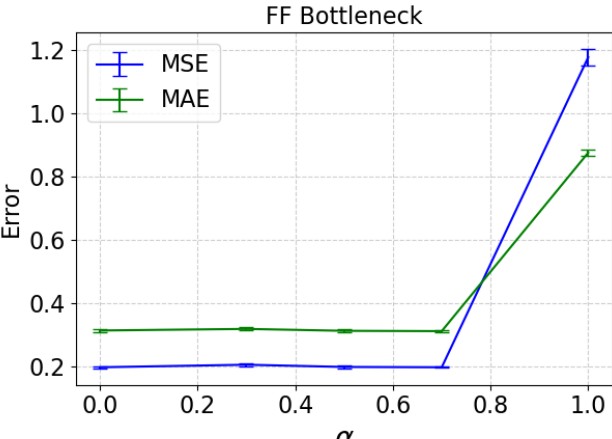

Figure 10: Performance of the Autoformer for different values of $\alpha$ in MSE and MAE. Note that this is a plot of the same results as given in Table 5.

Additionally, the CKA scores of the different models with the interpretable concepts (and other time features) are given in Figures 11, 12, and 13. Naturally, the CKA scores are the lowest in the setting $\alpha = 0$, and the scores from the bottleneck (`layer1`) increase over $\alpha$. Interestingly, the CKA scores from the bottleneck do not increase for higher values than $\alpha = 0.5$, although the scores of some other components do increase. This indicates that perfect similarity to some interpretable concepts (where the CKA score is equal to 1) may not be reached.

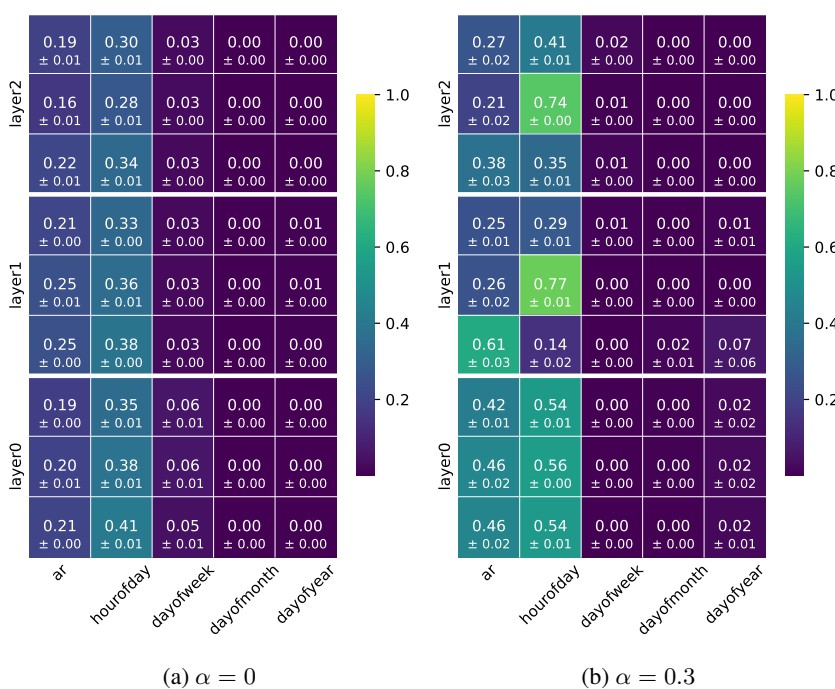

Figure 11: CKA scores of the feed-forward bottleneck Autoformer on electricity data for different values of hyperparameter $\alpha$. The scores are calculated using three batches of size 32 of the test data set.

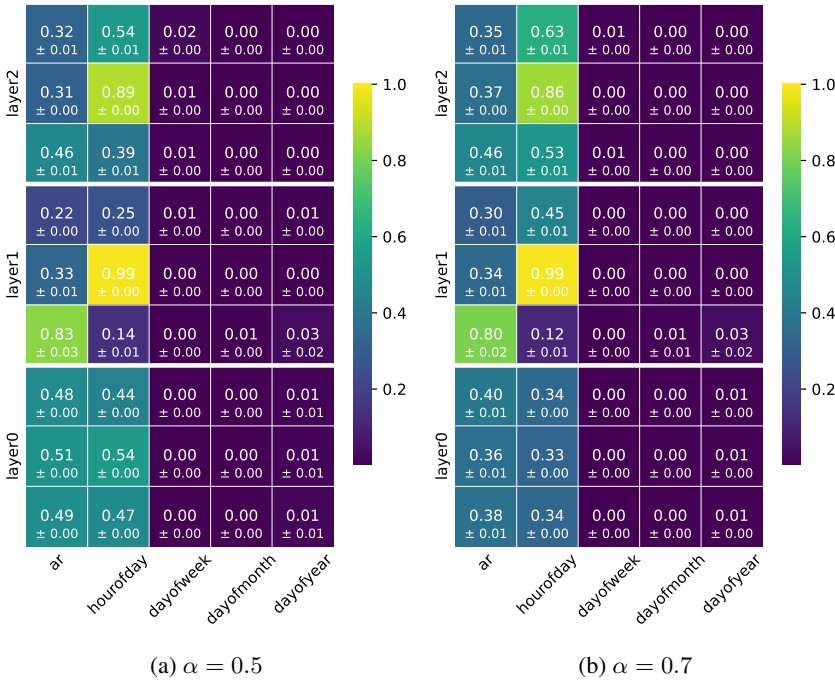

Figure 12: CKA scores of the feed-forward bottleneck Autoformer on electricity data for different values of hyperparameter $\alpha$. The scores are calculated using three batches of size 32 of the test data set.

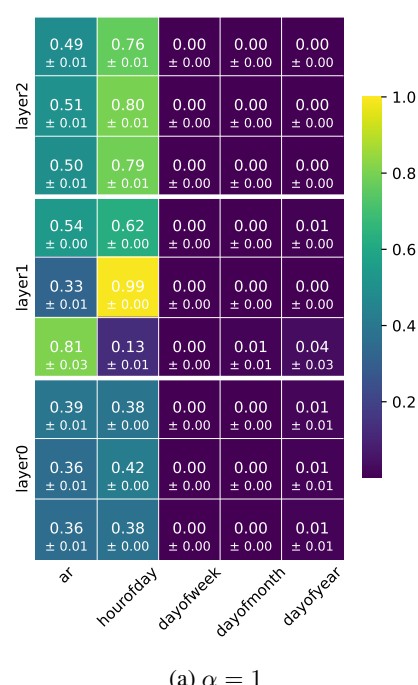

(a) $\alpha = 1$

Figure 13: CKA scores of the feed-forward bottleneck Autoformer on electricity data for hyperparameter $\alpha = 1$. The scores are calculated using three batches of size 32 of the test data set.

## G CKA ANALYSIS FOR MORE DATASETS

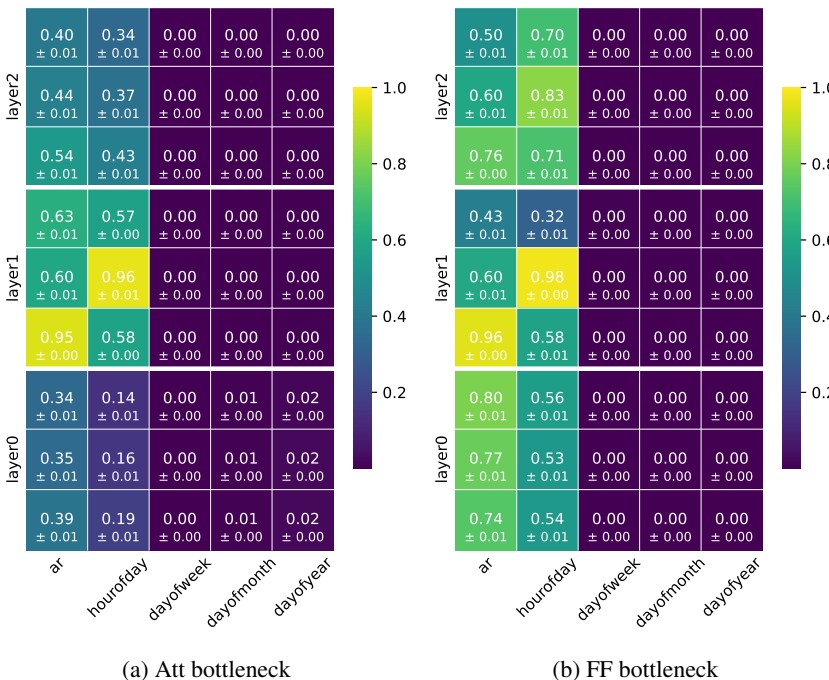

(a) Att bottleneck

(b) FF bottleneck

Figure 14: Traffic - CKA scores of the attention bottleneck Autoformer on traffic data. The scores are calculated using three batches of size 32 of the test data set.

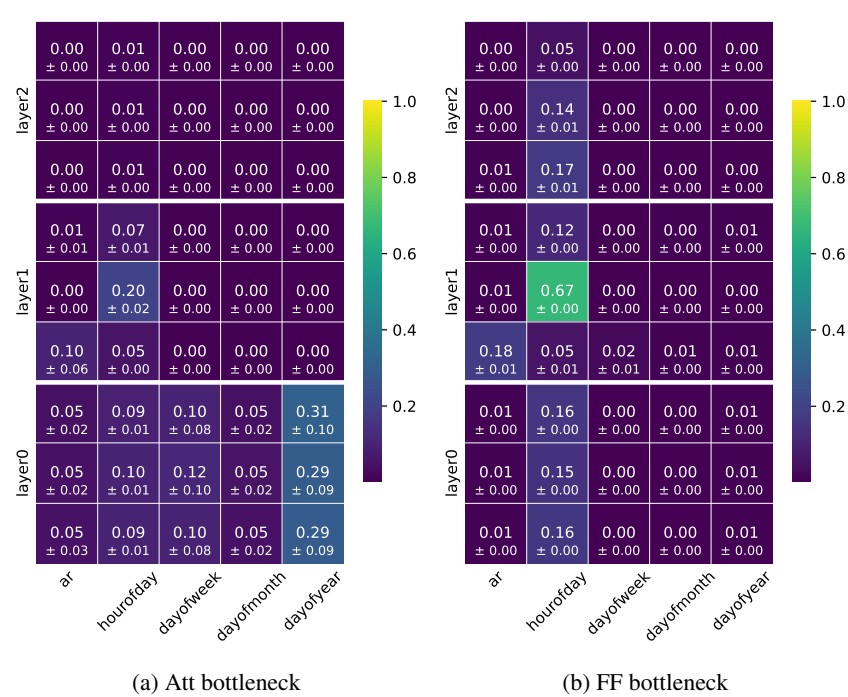

(a) Att bottleneck        (b) FF bottleneck

Figure 15: Weather - CKA scores of the attention bottleneck Autoformer on weather data. The scores are calculated using three batches of size 32 of the test data set.

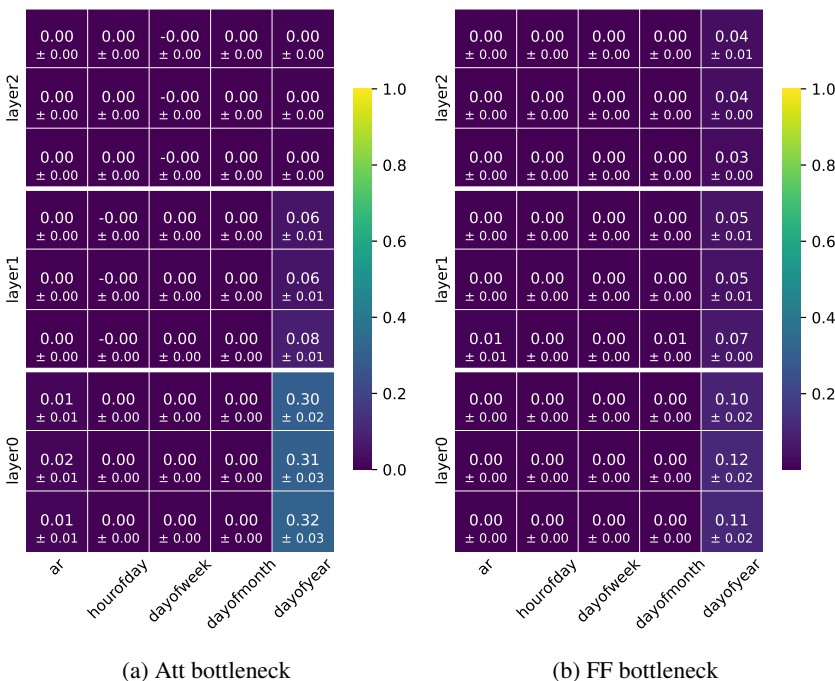

(a) Att bottleneck        (b) FF bottleneck

Figure 16: Illness - CKA scores of the attention bottleneck Autoformer on the illness data set. The scores are calculated using three batches of size 32 of the test data set.

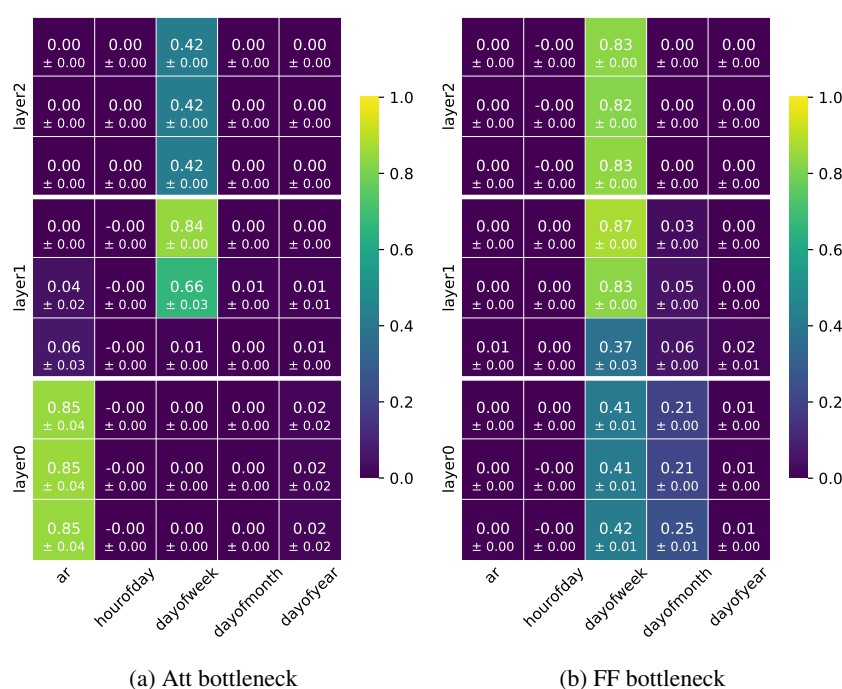

(a) Att bottleneck  (b) FF bottleneck

Figure 17: Exchange rate - CKA scores of the attention bottleneck Autoformer on the exchange rate data set. The scores are calculated using three batches of size 32 of the test data set.

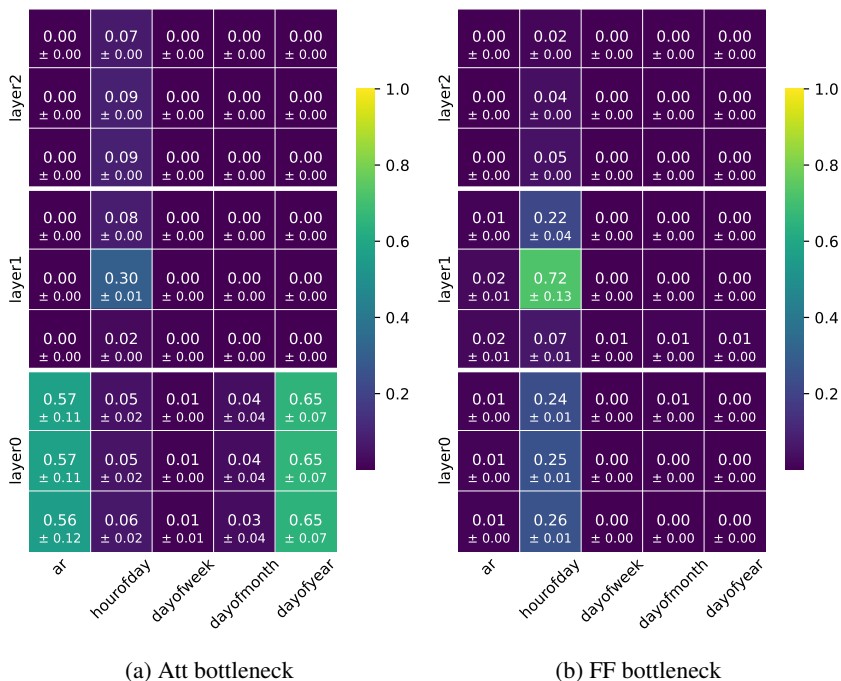

(a) Att bottleneck  (b) FF bottleneck

Figure 18: ETT - CKA scores of the attention bottleneck Autoformer on the ETT data set. The scores are calculated using three batches of size 32 of the test data set.

# H    APPLICATION OF FRAMEWORK TO VANILLA TRANSFORMER

To demonstrate the generality of the concept bottleneck framework, we apply it to an additional Transformer architecture, namely the *vanilla Transformer* (the original architecture from which all Transformer models, including all time series Transformers, are derived). We train it using the same six benchmark datasets and perform a similar, but less extensive, analysis as done for the Autoformer model.

## H.1    PERFORMANCE ANALYSIS

The performance of the vanilla Transformer model with and without bottleneck is given in Table 6. We train the bottleneck with a 'free' component (the side channel), i.e., with $c = 3$. The original Transformer paper does not provide scores for these benchmark forecasting datasets, therefore we cannot provide the 'Original' scores, as done for the Autoformer. The results show that the vanilla Transformer perform, unsurprisingly, worse than the Autoformer, and for most datasets also worse than the linear AR model. However, most relevant, for our purposes, is that across the datasets using a concept bottleneck does not hurt the overall performance of the vanilla Transformer.

Table 6: Performance of different vanilla Transformer models. For both metrics, it holds that a lower score indicates a better performance, where the best results are **bold**, and the second-best are underlined.

| | Att bottleneck | | FF bottleneck | | No bottleneck | | AR | |
|---|---|---|---|---|---|---|---|---|
| | MSE | MAE | MSE | MAE | MSE | MAE | MSE | MAE |
| Electricity | 0.275 | 0.371 | **0.268** | **0.362** | 0.275 | 0.371 | 0.497 | 0.522 |
| Traffic | 0.708 | 0.394 | 0.703 | 0.397 | 0.684 | **0.376** | **0.420** | 0.494 |
| Weather | 0.400 | 0.450 | 0.381 | 0.410 | 0.362 | 0.415 | **0.006** | **0.062** |
| Illness | 3.380 | 1.280 | 3.323 | 1.252 | 3.321 | 1.273 | **1.027** | **0.820** |
| Exchange rate | 0.675 | 0.642 | 0.677 | 0.633 | 0.694 | 0.662 | **0.082** | **0.230** |
| ETT | 0.230 | 0.328 | 0.185 | 0.299 | 0.166 | 0.294 | **0.034** | **0.117** |

## H.2    CKA ANALYSIS

After training the vanilla Transformer with the bottleneck framework, we evaluate the similarity of its hidden representations to the interpretable concepts using CKA, see Figure 19. Recall that CKA scores are defined in the range from 0 to 1, where 1 indicates perfect similarity. Both components in the two types of bottleneck show very high similarity to their target concept. Interestingly, the first component in the bottleneck (the AR concept) shows a higher similarity to the AR representations than the Autoformer (see Figure 3), presumably because the decomposition structure of the Autoformer hinders learning a linear function.

## H.3    COMPONENT VISUALIZATIONS

We visualize the contributions of each component in the bottleneck using the Decoder Lens method (Langedijk et al., 2023), see Figure 20. We obtain the output from each component individually by masking the other components with zero (close to the mean). Each component seems to provide similar contributions to the forecast as their respective counterpart in the Autoformer model. In particular, the first component (see Figure 20a) produces forecasts of correct seasonality and few irregularities, similar to the AR model. The second component (see Figure 20b) follows the 'hour-of-day' feature, and the free head (see Figure 20c) picks up on high-frequency data patterns.

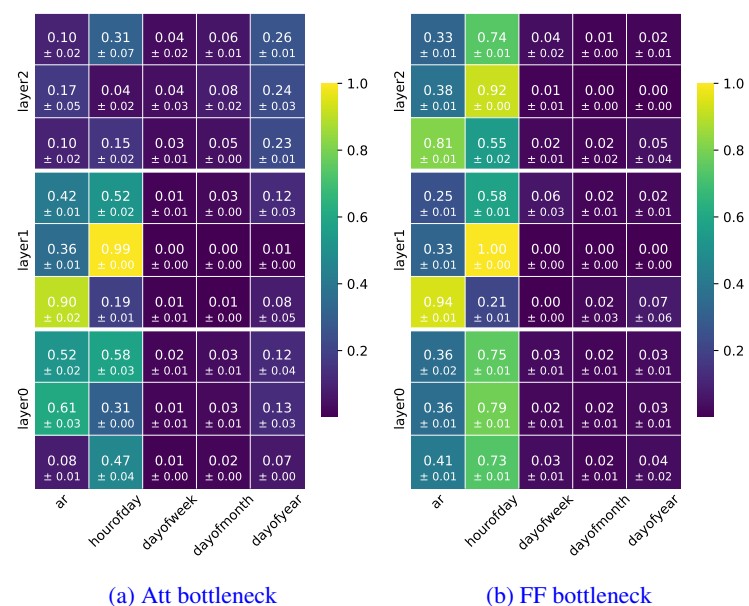

(a) Att bottleneck          (b) FF bottleneck

Figure 19: CKA scores of the vanilla Transformer's encoder (containing three heads per layer) from the attention and feed-forward bottleneck on the electricity dataset, where each score denotes the similarity of an individual component. The first component of `layer1` is trained to be similar to AR, and the second component to the 'hour-of-day' concept (lower and middle row in the figure, respectively). The scores are calculated using three batches of size 32 from the test data set.

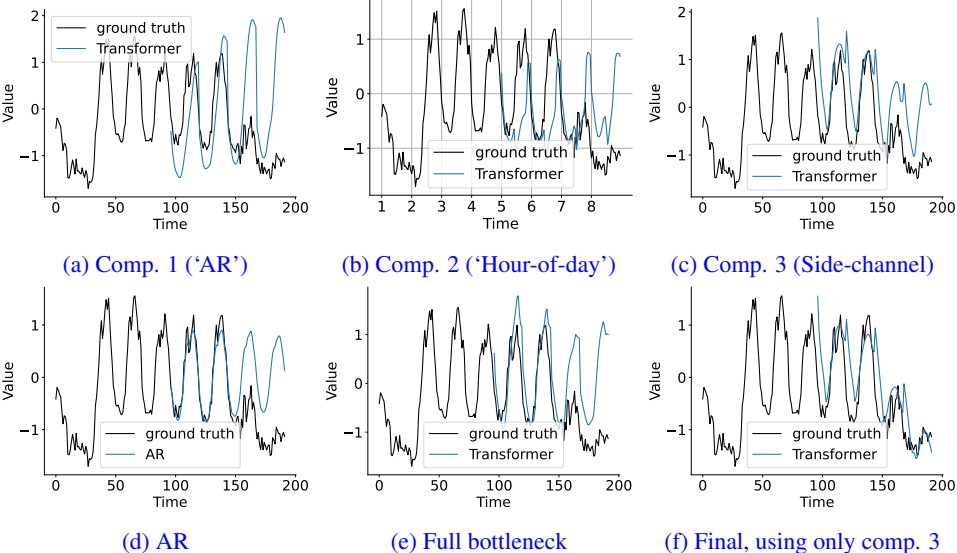

(a) Comp. 1 ('AR')     (b) Comp. 2 ('Hour-of-day')     (c) Comp. 3 (Side-channel)

(d) AR     (e) Full bottleneck     (f) Final, using only comp. 3

Figure 20: Vanilla Transformer forecasts from the components in the bottleneck layer (FF bottleneck on electricity data) in 4a, 4b and 4c. They are obtained by masking the other components with zero (the mean). The first half of the ground truth forms the input to the model. Note that the horizontal axes are the same across all figures, but Figure 4b contains a grid of days instead of numbered hours. Figure 20d shows the forecast made by the surrogate model AR; Figure 20e shows the forecast of the entire layer (i.e., all components together), and 20f shows the forecast of the final layer when only the third component is used in the bottleneck layer. Note the difference between Figures 4c and 20f, where we decode from the bottleneck and the final layer, respectively.

## H.4 INTERVENTION

We perform the intervention experiment in the same set-up as for the Autoformer model. That is, we delay the input timestamps with a fixed number of hours to obtain shifted timestamps, and perform an intervention in the bottleneck by substituting the activations based on the shifted time with the activations from the original time. We use a vanilla Transformer trained on the electricity dataset, and perform shifts of up to and including 23 hours. We compare the performance of the intervention with out-of-the-box performance of the same model on the shifted dataset. The results are shown in Figure 21. For both types of bottlenecks, the intervention performs best for all timeshifts, by keeping the error scores marginally close to the original performance (with no timeshift). This indicates that the model effectively learns to represent the 'hour-of-day' concept in the dedicated head, which is able to provide control over the model's behavior.

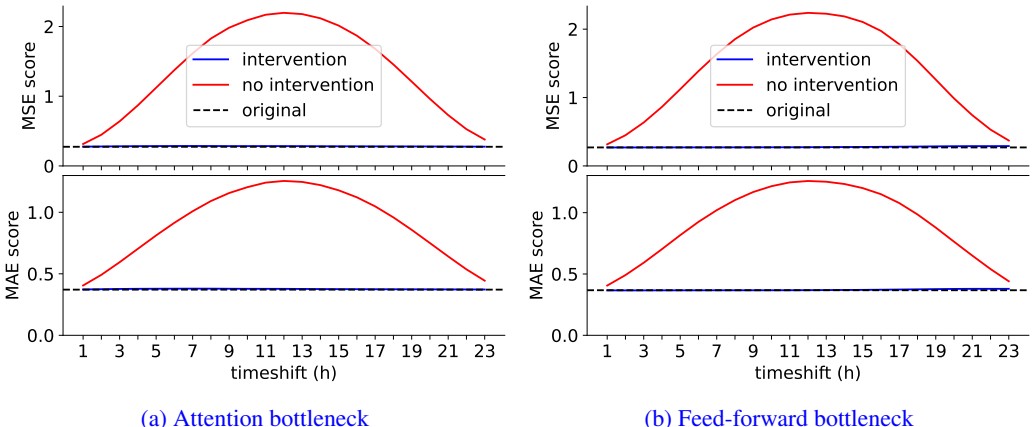

(a) Attention bottleneck    (b) Feed-forward bottleneck

Figure 21: Performance of the bottleneck vanilla Transformer on electricity data with shifted timestamps. The dashed line represents the performance of the same model on the original data, i.e., with no timeshift.

## H.5 CONCLUSION

By repeating the set of experiments for the vanilla Transformer model, we provided further evidence for the generality of the concept bottleneck framework. In particular, we showed that the framework can be applied to the vanilla Transformer model, without having any significant impact on the overall model performance, while providing improved interpretability.

# I SYNTHETIC DATA

To increase the understanding of how the concepts in the bottleneck can be leveraged, we train the model on a synthetic dataset.

## I.1 DATASET

We generate a synthetic time series as the sum of different functions. In particular, the dataset is generated using the function $f_{Total}$ with time $t$ as follows:

$$f_{Total}(t) = f_1(t) + f_2(t) + f_3(t),$$

where:

$$f_1(t) = \sin(2\pi t),$$
$$f_2(t) = \frac{1}{2}\sin(4\pi t + \frac{\pi}{4}),$$
$$f_3(t) = \frac{1}{4}\sin(6\pi t + \frac{\pi}{2}) + \epsilon_t.$$

Note that all functions $f_1, f_2$ and $f_3$ follow a periodic structure, and $f_3$ contains random noise $\epsilon$ from a normal distribution with standard deviation of 0.2. See Figure 22 for a visualization of the functions.

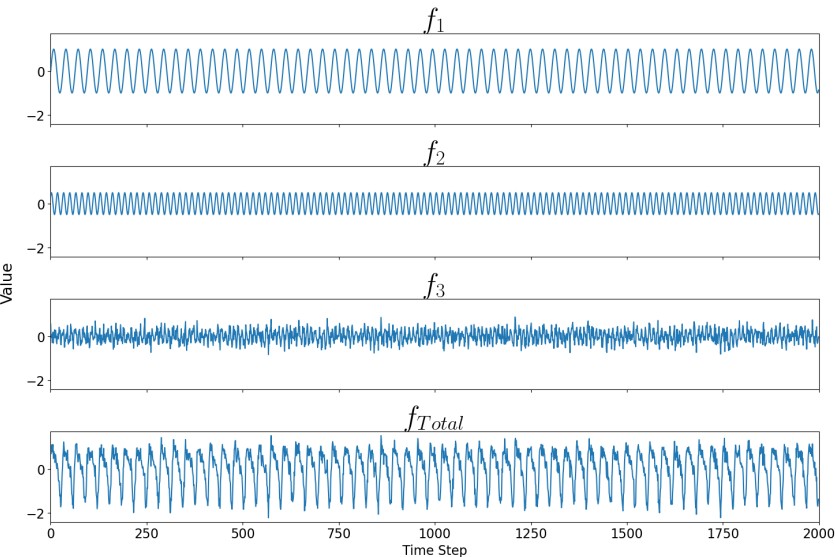

Figure 22: The synthetic time series dataset.

## I.2 EXPERIMENT AND RESULTS

We train an Autoformer model on the synthetic dataset using the concept bottleneck framework. Each concept in the bottleneck is defined as one of the underlying functions (i.e., $f_1$, $f_2$ or $f_3$), for which the ground-truth is known by construction. The model contains three encoder layers, with three attention heads per layer. We apply the bottleneck to the attention heads of the second encoder layer. Additionally, we train the bottleneck using different values for hyperparameter $\alpha$, which controls the weight of the CKA loss in the total loss function (see Section 3.1).

As expected, we find for all values $\alpha < 1$ that the model is able to forecast the dataset well, see Figure 23. Note that a low forecasting error cannot be expected for $\alpha = 1$, because in this edge case the loss function does not contain any forecasting error. Remarkably, for all other cases, the performance of the Autoformer seems to improve as $\alpha$ increases. This suggests that properly chosen concepts improve the performance of the model, at least when the ground-truth underlying functions

are known. It should be noted that the standard deviation is higher for all $\alpha > 0$, which indicates that initialization of the parameters is important when learning the bottleneck. Additionally, visualizations of the predictions are given in Figure 24.

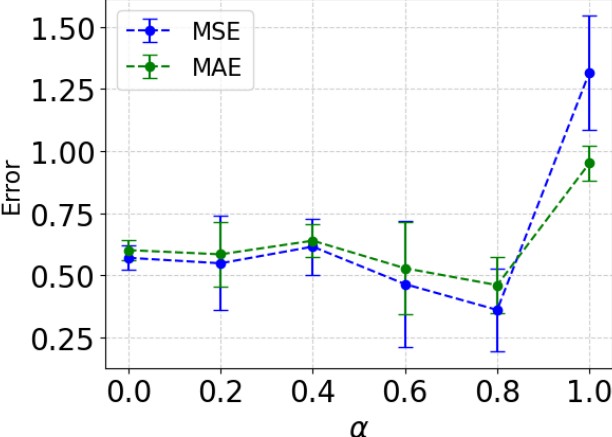

Figure 23: Performance on the synthetic dataset for different values of $\alpha$, using an Autoformer with attention bottleneck. For both metrics, it holds that a lower score indicates a better performance. The standard deviation is provided over 5 different seeds.

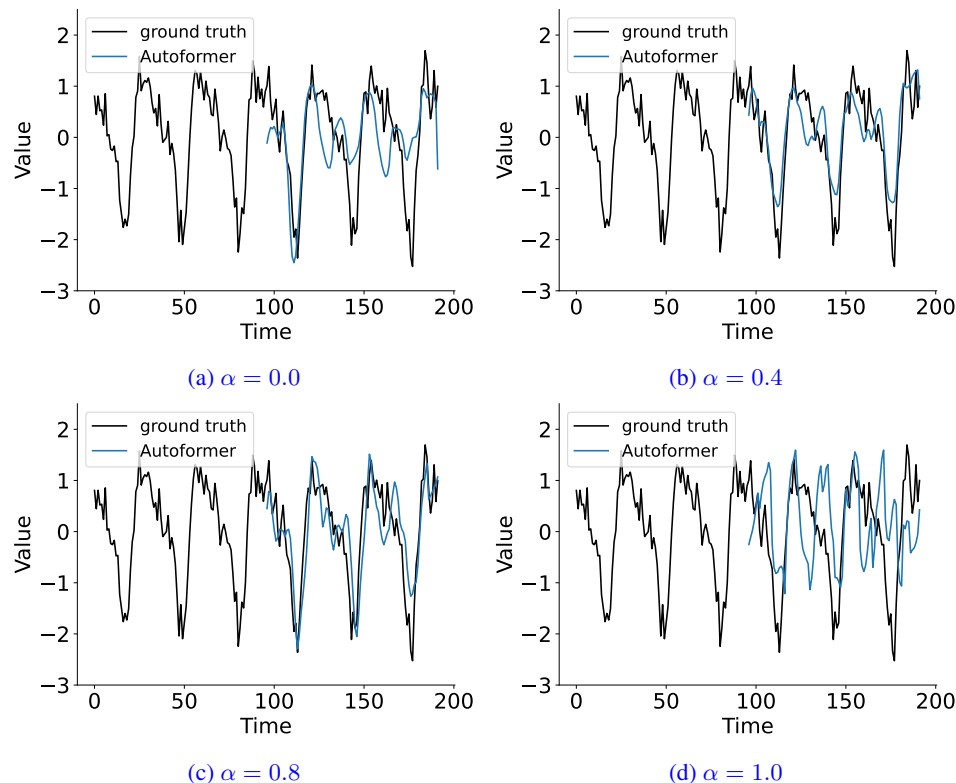

Figure 24: Predictions of the Autoformer model on a sample from the test dataset. The Autoformer is trained with an attention bottleneck using different values of hyperparameter $\alpha$ and the same seed.

Additionally, the different values of hyperparameter $\alpha$ show clearly how the different concepts are leveraged by the model, see Figure 25. The figure shows the similarity scores between the attention heads and the different underlying functions of the dataset. Without the CKA loss, at $\alpha = 0$, the

different heads in layer1 of the model do not show high similarity to their respective concepts, i.e., functions. Instead, all heads have a high similarity to concept $f_2$. This is different for higher values of $\alpha$, where the different heads show higher similarity to their respective concepts. Note that the third concept $f_3$ cannot be perfectly learned by the model because of the random noise component.

All in all, these results show that a higher value for $\alpha$, which is equivalent to a higher weight of the CKA loss in the total loss function, results in more similarity of the bottleneck components to their respective concepts, as expected.

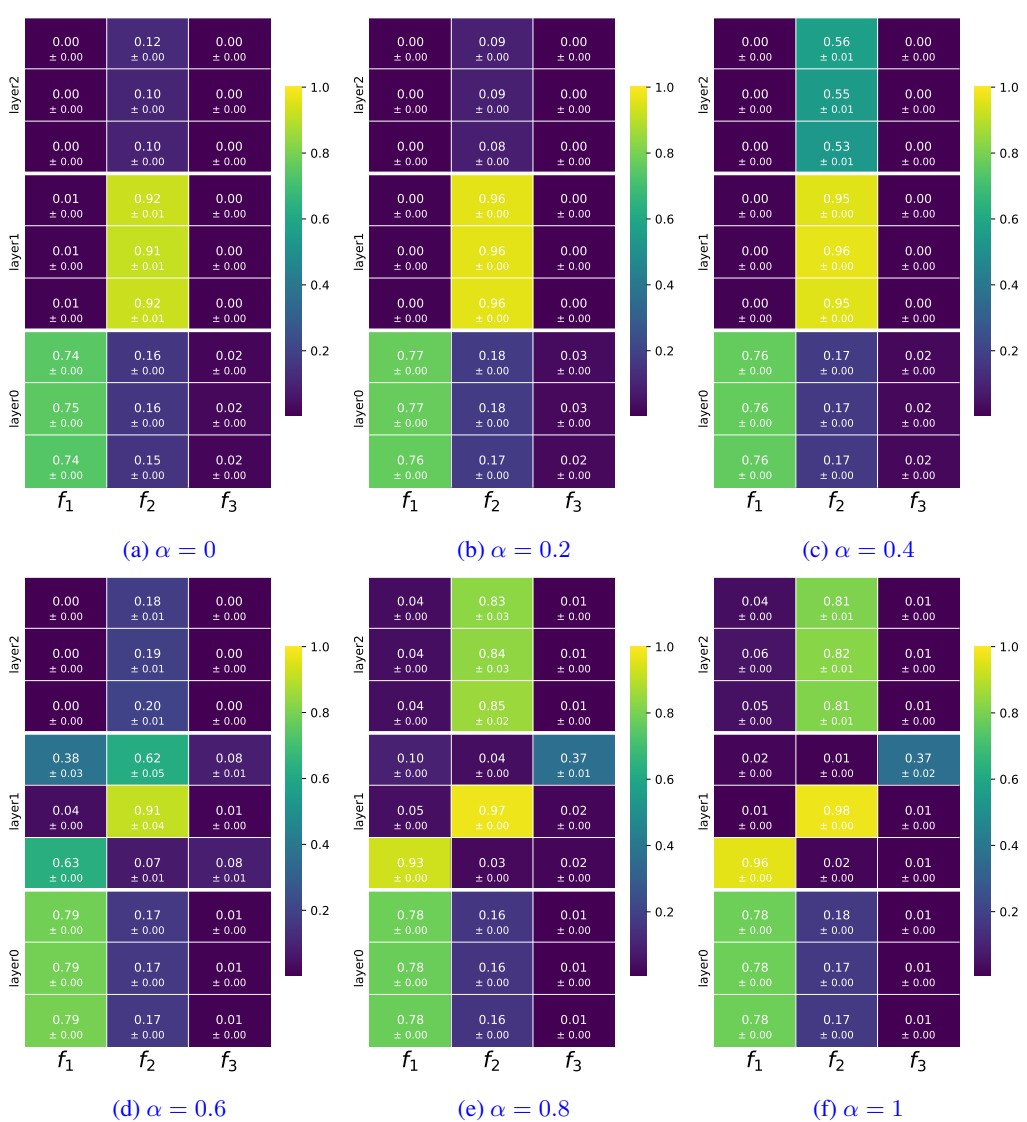

Figure 25: CKA scores of the attention bottleneck Autoformer on synthetic data for different values of hyperparameter $\alpha$. The scores are calculated using three batches of size 32 of the test data set.

## J   EFFECT OF AR AS SURROGATE MODEL

Interestingly, the AR model outperforms the Autoformer for some datasets (see Table 1). This raises the question whether the AR surrogate model makes up for any loss in performance introduced by the concept bottleneck.

To test this, we train an Autoformer without the AR concept. Specifically, we include the time concept and a free component in the feed-forward bottleneck. Here, the free component refers to a component in the bottleneck that is not included in the CKA loss (see Section 3.2).

The performance on the electricity data for this model is (MSE: 0.206, MAE: 0.321), which is seemingly identical to the original performance of (MSE: 0.207, MAE: 0.320). This suggests that it is not the AR head that makes up for the loss in performance. The CKA plots, see Figure 26, verify that there is no component in the minimal set-up (without AR) that is very similar to the AR model, unlike in the original set-up. So, these results show that the AR model does not add performance to the bottleneck model, merely interpretability.

Additionally, we refer the reader to Appendix I, where we perform more experiments on training the bottleneck without the AR surrogate model.

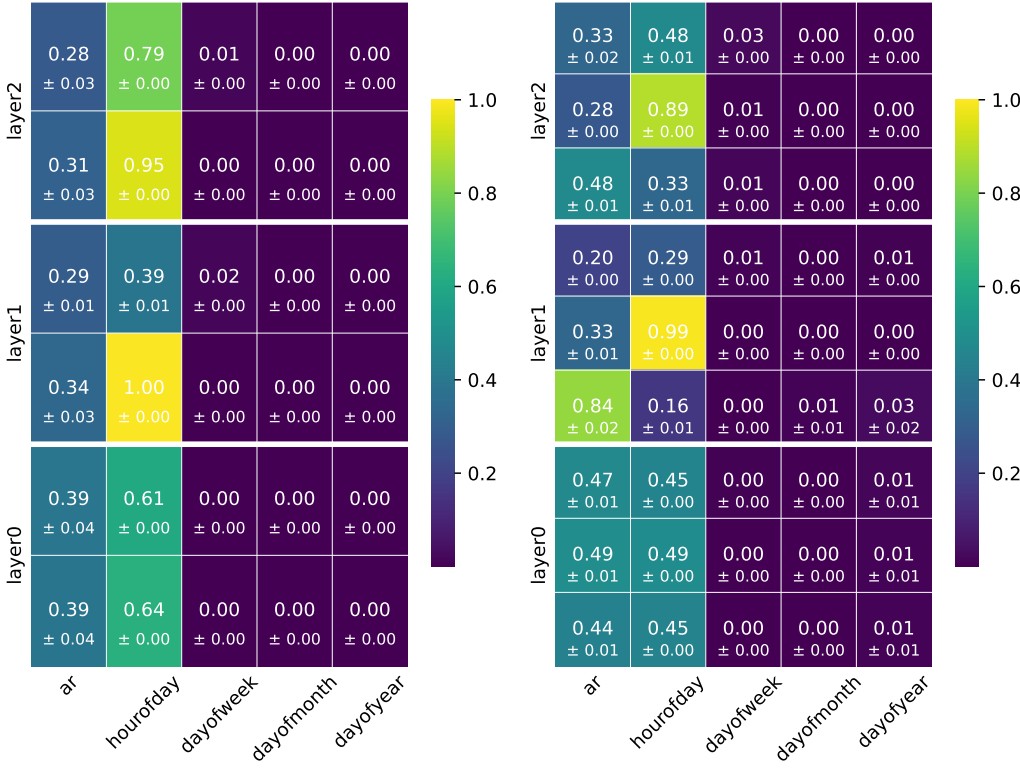

(a) Without AR (MSE: 0.206, MAE: 0.321)    (b) With AR (MSE: 0.207, MAE: 0.320)

Figure 26: CKA plots of two Autoformer models with feed-forward bottlenecks. The model in 26a is trained without AR in the bottleneck, while the model in 26b is trained with AR. Note that the upper component in `layer1` is the free component in both plots.