# OpenReview forum: "Enforcing Interpretability in Time Series Transformers: A Concept Bottleneck Framework"
_ICLR.cc/2025/Conference — Submitted to ICLR 2025_

### Official Review · Reviewer_M89B · 2024-10-30

**Soundness:** 3
**Presentation:** 3
**Contribution:** 3
**Rating:** 6
**Confidence:** 3

**Summary:**

This paper proposes a framework to enforce interpretability in time-series forecasting Transformers by adapting the Autoformer model with a concept bottleneck approach. The framework aligns the model’s representations with interpretable concepts, such as a surrogate AR model and time-based features, using Centered Kernel Alignment (CKA). This structure aims to make parts of the Autoformer model more transparent, allowing practitioners to interpret the model’s reasoning and make targeted interventions if needed. The paper demonstrates that the proposed framework maintains interpretability with a minimal performance trade-off across six time-series datasets.

**Strengths:**

1. Novel Interpretability Framework: This work contributes a new direction in Transformer interpretability by combining Concept Bottleneck Models (CBMs) with the Autoformer, explicitly aligning model representations with interpretable concepts.

2. Few Performance Trade-offs with Useful Intervention Property: Table 1 shows that while the Autoformer with bottlenecks generally has a slight performance trade-off, the interpretability improvements may be valuable in settings where transparency is essential. Additionally, the “Intervention” experiment (Lines 480-485) demonstrates a practical application of the framework, where a temporal shift intervention shows the model’s adaptability to new data distributions, a useful feature in evolving environments.

4. Potential Balance of Interpretability and Complexity: By modifying only a single layer to incorporate the concept bottleneck and aligning some heads with interpretable concepts, the framework achieves interpretability without overhauling the Transformer architecture. This approach makes the Autoformer’s components "easily intervenable," according to the authors, providing a possible solution for practitioners needing complex forecasting models with interpretable checkpoints.

**Weaknesses:**

1. Limitations in Granular Interpretability: CKA encourages global alignment of the bottleneck representations with the predefined concepts, which may not capture fine-grained temporal patterns that are essential in many time-series applications. In the CKA analysis (Figure 3), alignment scores reflect similarity with concepts on a broad level but do not offer insights at specific time intervals or for anomalies. This setup could limit interpretability for users who need detailed, time-specific insights. Extending the interpretability framework to capture these localized patterns would make the model’s insights more actionable.

2. Interpretability Evaluation Metrics: The interpretability evaluation relies mainly on CKA scores and qualitative visualizations. Although CKA scores indicate alignment between model representations and interpretable concepts, they do not provide a full measure of "practical interpretability" from an end-user perspective. Incorporating metrics that measure interpretability in terms of clarity or usefulness for decision-making could make the framework’s impact clearer and more valuable.

3. Applicability Across Different Models: Although the framework is applied to the Autoformer model, extending it to other more performant Transformer-based time-series models would confirm its generalizability. While the authors mention this as a possible future direction (Lines 530-531), this limits the scientific contribution of the work.

4. Model diagrams (Figure 1 and 2) and the CKA scores (Fig 3) could be presented more clearly. They often require frequent referrals back to the text and legend.

**Questions:**

1. Justification for Concept Bottleneck over Post-Hoc Methods: Could you clarify the specific advantages of using the concept bottleneck framework over post-hoc interpretability methods like SHAP, LIME, or attention-based visualizations in this time-series context? An expanded discussion would help in understanding the unique benefits of your approach for interpretability.

---

> ### Author Response · Authors · 2024-11-19
> **Justification concept bottleneck models over post-hoc interpretability**
>
> Thank you for insightful comments and the extensive review. We appreciate your question on the justification for concept bottleneck models over post-hoc methods.
>
> Our motivation for the concept bottleneck framework over post-hoc interpretability is two-fold. Firstly, the post-hoc interpretability methods are notorious for not being faithful to the model’s mechanisms. By enforcing interpretability already at training time, we attempt to overcome this and make the model interpretable by design.
>
> Secondly, post-hoc interpretability turns out to be difficult when trying to localize specific features (or concepts) within the model. For example, we can assume any trained Autoformer model should have learned some concept of time, yet, the CKA scores do not show any specific head to have a high similarity to this (see Figure 10a in Appendix F). In practice, these high-level concepts seem to be distributed amongst different model components. (In some sense, our approach is thus complementary to the current popular idea of using Sparse Auto-Encoders (SAE) for interpretability: while SAE are often used to deal with individual neurons having different functions (i.e., “polysemanticity”), our approach manages to reduce the distributed nature of representations).
>
> The highly distributed representation of concepts also makes it difficult to compare different model architectures, and to find out what they exactly picked up from the data. Additionally, locality of concepts can be beneficial if these concepts are important to understand and gain control over the model’s internal mechanisms, so that an intervention can be done (as we show in our intervention experiment).
>
> We see the potential for actionable results that indicate how the input should change to obtain a different outcome. And we agree with you that the model’s insights would be more actionable when localized patterns are captured. However, that is outside the scope of our current paper, which is rather about understanding what a model learns from the data and how we can influence it.

---

### Official Review · Reviewer_ztC3 · 2024-11-03

**Soundness:** 3
**Presentation:** 3
**Contribution:** 3
**Rating:** 6
**Confidence:** 4

**Summary:**

The paper proposes a time series Transformer model to be more interpretable with concept bottlenecks using time features and simple autoregressive models as interpretable concepts.
* A training framework encouraging the similarity between transformer representations and pre-defined interpretable concepts using CKA.
* Application of Autoformer on the model was made and its performance was evaluated on 6 benchmark datasets.
* Demonstrate the capability of model intervention in case of temporal shifts.
* Extensive interpretation analysis supported by the visualization technique.

**Strengths:**

* Novel application of CBM to time series
* Creative use of CKA for concept alignment
* Integration with Autoformer architecture
* Novel intervention capabilities

* Comprehensive experiments on 6 datasets
* Detailed ablation studies
* Visualization of learned concepts
* Intervention demonstration

* Comparable performance to baseline
* Interpretable predictions
* Intervention capabilities for temporal shifts
* Domain-agnostic approach

**Weaknesses:**

* Single model architecture (Autoformer)
* Further research is needed to apply CBM to other types of predictive models
* Selection of interpretable concepts relies on heuristics
* Limited analysis of statistical significance
* No comparison to other interpretability methods

* Potential information leakage not fully addressed
* Limited analysis of concept quality
* No theoretical guarantees
* Trade-offs not fully explored


* In Table 1, the simple AR model outperforms the Autoformer with bottleneck in 4 out of 6 datasets. Since the AR model is inherently interpretable, these results may suggest that the proposed method is less effective than expected. The authors could consider adding more complex datasets to strengthen the experimental evaluation.

Suggestions for Improvement
* Compare with other interpretability methods; Add user studies with domain experts; Provide more complex intervention scenarios; Test on longer sequences
* Study concept quality metrics; Analyze computational overhead; Evaluate statistical significance; Investigate scaling properties
* Test with other transformer architectures; Explore more complex concepts; Add theoretical guarantees

**Questions:**

* Is there a qualitative or quantitative comparison of the proposed method with other XAI techniques?

* The results of the intervention experiment are intriguing, but the purpose of this experiment remains somewhat unclear. Could the authors provide a more detailed analysis, discussion, or examples to clarify this?

* How were the specific interpretable concepts (AR model and time features) chosen? Were other concepts considered?

* How do you validate that the learned concepts are truly interpretable and meaningful? Have you conducted any user studies with domain experts?

* Why was Autoformer specifically chosen as the base architecture?
Would the approach work similarly with other transformer variants?

* What is the impact of bottleneck location on performance and interpretability? Was there a systematic study of different locations?

* How sensitive is the training to the CKA loss weight α? Are there guidelines for selecting this parameter?

* What is the computational overhead of the bottleneck compared to standard Autoformer? How does this scale with sequence length?

* How do you quantitatively evaluate the quality of interpretations? Are there metrics beyond CKA scores?

* How generalizable is the intervention approach to other types of shifts? What are the limitations?

---

> ### Author Response · Authors · 2024-11-19
>
> Thanks for the useful comments, and for your appreciation of the novelty of our approach. We will briefly try to answer your questions:
>
> * Comparison with other XAI techniques
> Comparing the interpretability of our framework with other models is not straightforward, because other interpretable time series models are conceptually orthogonal i.e., they often explain their predictions in terms of features of the input data, for example by perturbations [1] or attention scores [2, 3]. To the best of our knowledge, our work is the first to enforce a transformer to learn pre-determined concepts from the data, and use those in the down-stream task. Therefore, we cannot compare our interpretability of these concepts with another method. Essentially, we cannot use another method to enforce the model to learn a concept in a specific head, but our own (see also our response to the reviewer RmBn).
> * Purpose intervention experiment
> The aim of the intervention is two-fold: first to show a possible real-world application of the concept bottleneck framework, and secondly to verify the localization of concepts. By showing the timestamps intervention works, we verify that the concept of timestamps is indeed located in the intended head.
> * Choice of interpretable concepts
> The reason for the employed interpretable concepts (i.e., AR and the  timestamps) is mainly due to the fact that they are domain-independent, and therefore should work well across all datasets, in a respective manner. One could also consider potentially more sophisticated concepts, such as holidays, or special events, however we wanted the concepts to be representative and insightful for all datasets.
> * Validation of interpretability
> Our validation of the interpretable concepts is done with the intervention experiment. We do not conduct any user studies, even though we agree that these could be very valuable, because whether explanations are meaningful to the end user is an entire field of its own. We merely focus on providing a technical framework.
> * Choice of Autoformer architecture
> The Autoformer architecture was chosen because it is a well-studied  prominent Transformer model for time series forecasting. The framework architecture can be applied just as well to other Transformer architectures. We are in fact planning to apply it to other models in follow-up experiments, as part of our research agenda.
> * Bottleneck location
> We compared two locations for the bottleneck: in the attention vs. the feed-forward component, both in the encoder. Overall, we find that the feed-forward bottleneck performs slightly better for most datasets (see Table 1). We focus on modelling the encoding of interpretable concepts, so choosing a bottleneck location in the decoder does not align with  that idea.
> * Sensitivity to the alpha hyper-parameter
> Results about sensitivity to the hyperparameter alpha are given in Appendix F. The training does not seem to be overly sensitive to the hyperparameter: That is, even with a relatively high importance to the CKA loss (high value for alpha), the model is able to achieve low forecasting errors.
> * Computation overhead
> The extra computation from our method arises from calculating the CKA loss, which depends on the CKA score. This score (using a linear kernel) has quadratic complexity in the sequence length $n$, while the Autoformer has $n \text{ log } n$ complexity.
> * Quantitative evaluation of interpretations
> Quantitatively evaluating the interpretations is tricky, because there is no ground truth. While we use the CKA scores to evaluate, we also make use of the intervention to illustrate that the timestamps concept is indeed encoded by the head with the high CKA score to time.
> * Generalizability of intervention to other types of shift
> The limitation to the intervention is that one should have access to the shifted data, and know in which concepts the shift has occurred, so that the hidden representations from these concepts can be replaced. While this can be very well applied to a temporal  shift, the notion of intervention is not limited to it.
>
> [1] Enguehard, J. (2023). Learning Perturbations to Explain Time Series Predictions. ICML 2023
>
> [2] Davies, H. J., Monsen, J., & Mandic, D. P. (2024). Interpretable Pre-Trained Transformers for Heart Time-Series Data. arXiv preprint arXiv:2407.20775.
>
> [3] Temporal fusion transformers for interpretable multi-horizon time series forecasting. International Journal of Forecasting 2021

---

> > ### Comment · Reviewer_ztC3 · 2024-11-26
> >
> > Overall, the authors provided clear responses while being transparent about current limitations and future research directions.
> > - Their approach is unique in enforcing pre-determined concept learning in transformers
> > - They acknowledge the difficulty in direct comparison with other XAI methods due to different conceptual approaches
> > - Intervention experiments serve dual purposes: demonstrating real-world applications and verifying concept localization
> > - Choice of AR and timestamps as concepts was based on domain-independence
> > - Autoformer architecture selection was justified by its established performance
> >
> > Limitation:
> > - No user studies with domain experts
> > - Limited to specific types of concept shifts
> > - Need for access to shifted data for interventions
> > - Focus on technical framework rather than user interpretation

---

### Official Review · Reviewer_RmBn · 2024-11-03

**Soundness:** 3
**Presentation:** 2
**Contribution:** 2
**Rating:** 5
**Confidence:** 4

**Summary:**

The authors develop a concept bottleneck model for time-series forecasting with the objective of improving interpretability. Concept bottleneck models are a pre-existing approach to interpretability whereby the model aims to predict a set of concepts first, and then only uses the predicted concepts for the final forecast.

Starting from the Autoformer architecture, the authors introduce two types of bottlenecks (an autoregressive forecast and a time-of-day prediction). To ensure all information passes through the bottlecneck, they then ablate the residual connections. Finally the training loss is an interpolation of the standard loss + a score based on the similarity score CKA of the model’s representations and interpretable concepts.

**Strengths:**

The strengths are as follows:

- The paper is well-written, and the idea is expressed clearly.
- The authors achieve what they set out to do: their model functions at the intended task.

**Weaknesses:**

The weaknesses of the paper are as follows:

- In reviewing the performance results in Table 1, as the authors themselves acknowledge there is no significant improvement in performance (as is to be expected given the algorithm, this is of course not an issue). The paper however lacks a comparative analysis of this interpretability against other methods that also offer interpretable time-series prediction: does their approach outperform others in that space?

- Although the concept is intriguing, it feels somewhat derivative, essentially applying concept bottlenecks to time-series forecasting. One immediate concern is the relative lack of novelty. This may not be a significant issue if there were more extensive analysis of the components in their approach, yet the exploration remains somewhat limited. Specifically, other proxy tasks for the interpretable concepts could have been explored, as well as other components (e.g. bottleneck location, similarity metric used, transformer models...).

- The authors note that the AR model outperforms other approaches. This finding is not unexpected given prior work (e.g., [1]), but further analysis is warranted. The key unanswered question, in my view, is how much of the absence of performance degradation is due to the strong proxy task provided by AR (i.e. is their model performing as well as the unaltered baseline only due to the strong signal provided by the AR subtask).

- The dataset selection is somewhat limited. The authors mention that the time-series analyzed in this study had strong linear characteristics, which likely explains the AR model's performance. This could motivate the use of more complex datasets to verify if the findings hold more broadly.


References
[1] FreDo: Frequency Domain-based Long-Term Time Series Forecasting.

**Questions:**

Please refer to the weaknesses section above for questions.

---

> ### Author Response · Authors · 2024-11-19
>
> Many thanks for the useful comments! We try to address the 4 potential weaknesses you mention, and hope to convince you to convince the scores somewhat.
>
> ### W1: Comparative analysis of interpretability
>
> We agree that comparing the interpretability of our framework with other approaches would be very interesting, but doing such a comparison is tricky. The main reason is that other interpretable time series models typically explain their predictions in terms of features of the input data, for example by perturbations [1] or attention scores [2, 3]. To the best of our knowledge, our work is the first to enforce a transformer to learn pre-determined concepts from the data*, and use those in the down-stream task.
>
> In our study, we concluded that the best comparison is between a Transformer model *before* and *after* applying the “enforcing” interventions from our framework. For these results, we refer to Figure 10a in Appendix F. The visualization shows that the model components show limited similarity to AR and the different time concepts, whereas the similarity increases when applying our framework.
>
> ### W2: More extensive research analysis
>
> It is, of course, difficult to argue about how ‘derivative’ new work is. We just note that there has been much interest in both concept bottleneck models and time series interpretability (as we review in the paper), but that the combination we propose – which builds on insights from a diversity of subfields (including mechanistic interpretability and CKA) has not been proposed before. We have tried to do all the necessary analyses to support our claims!
>
> ### W3: Impact AR in preventing model degradation
>
> Many thanks for raising this interesting question: does the AR surrogate model make up for any loss in performance introduced by the concept bottleneck? We did, in fact, perform an experiment that partially answers it. We trained an Autoformer without the AR concept, but with the time concept and a free head. The performance on the electricity data for this model is (MSE: 0.206, MAE: 0.321), which is seemingly identical to the original performance of (MSE: 0.207, MAE: 0.320). This suggests that it is not the AR head that makes up for the loss in performance. When looking at the CKA plots, we find that the free head in the minimal set-up (without AR) has less similarity to the time concept than in the original set-up, indicating that it learns less similar representations than before. So, instead of adding performance to the bottleneck model, we believe these results show that the AR model just adds interpretability, which is in line with the claims of our paper.
>
> ### W4: More complex datasets
>
> We agree that the use of more datasets would be interesting, but would argue that they are not at this stage necessary to test the robustness of the framework. We did not cherry-pick our datasets; rather, we used the full suite of commonly used datasets from the recent time series literature (including those in the original [celebrated] Autoformer paper, including datasets [Traffic and Electricity], for which the Autoformer model outperforms AR). In fact, there is an ongoing discussion about the application of Transformers for time series (see our response to reviewer J9av).
>
> [1] Enguehard, J. (2023). Learning Perturbations to Explain Time Series Predictions.ICML 2023.
>
> [2] Temporal fusion transformers for interpretable multi-horizon time series forecasting, International Journal of Forecasting 2021
>
> [3] Davies, H. J., Monsen, J., & Mandic, D. P. (2024). Interpretable Pre-Trained Transformers for Heart Time-Series Data. arXiv preprint arXiv:2407.20775.
>
> *Note that there is work on concept-based anomaly detection for time series (Ferfoglia, I., Saveri, G., Nenzi, L., & Bortolussi, L. (2024). ECATS: Explainable-by-design concept-based anomaly detection for time series. ArXiv, abs/2405.10608.) However, this work represents concepts as Signal Temporal Logic formulae, such as, “the temperature should never exceed a certain threshold for more than a specified duration”. In contrast, by ‘concepts’, we mean high-level features from the time series data.

---

> ### Comment · Reviewer_RmBn · 2024-11-28
> **Response to authors**
>
> I have read the authors' response and thank them for taking the time to address my points.
>
> A few comments:
>
> # W2:
> Taken from the author's response to my point:
> > We have tried to do all the necessary analyses to support our claims!
>
> Taken from my review:
> > Specifically, other proxy tasks for the interpretable concepts could have been explored, as well as other components (e.g. bottleneck location, similarity metric used, transformer models...).
>
> To be clear, I have no issues with the work being potentially somewhat derivative, as I state. My issues is that among the free variables of the problem, not many are sufficiently explored. Does a single, arguably widely recognized transformer paper from 2021 consitute a sufficient exploration into the possible transformer backbones that could be used? The same goes for the other components, which my review was an invitation to explore.
>
> # W4:
> Yes, I agree that there is extensive debate about the validity of transformers for time-series forecasting, and arguably part of that debate stems from the fact that perhaps the datasets we commonly evaluate such models on are too limited to draw robust conclusions. The FreDO paper that I mention, e.g. shows that since such datasets commonly have a strong frequency component, a non-parametric model is already good at forecasting on them. This does not mean that such a baseline would hold for all conceivable datasets, hence my question.
>
> Respectfully, I do not feel the authors have answered my point when they mention that they have used the datasets commonly found in the literature. They set out to show that their approach brings value. I mention that it may be caused by dataset bias, and wonder if such performance would remain in some datasets for which AR is not such a strong candidate. Merely stating that the common datasets are indeed the ones that other papers tend to use does not, in my opinion, address this. This ties into my above point (W2) about insufficient exploration of the hyper-parameter/problem space.
>
>
> # W1
> I agree with the reviewer's comments about the relatively novel nature of the problem, and their justification.
>
> # W3
> This experiment seems to go in the right direction towards adressing this point. Could I just ask the authors to explain in more detail what they mean by this:
> > We trained an Autoformer without the AR concept, but with the time concept and a free head.
> (I want to be sure that I understand correctly the procedure here, esp. the free head).

---

> > ### Author Response · Authors · 2024-11-28
> >
> > Thank you for reading our response, and we appreciate your comments.
> >
> > Perhaps we did not state this clear enough, but we have accepted your invitation to explore. We have performed deeper explorations since your original review (announced by global comments, but not mentioned in the personal comment).
> >
> > ## W2
> > Specifically, to address W2, we apply the framework to the Vanilla Transformer, to show the generality of the framework (Appendix H). Our framework therefore does not depend on a single transformer paper.
> >
> > ## W4
> > Furthermore, we understand your worry for dataset bias, and agree that this would be problematic. To show there is no dataset bias (W4), we apply the framework to a synthetic dataset (Appendix I). In this experiment, we do not train the bottleneck with the AR concept. Instead, we use the underlying functions of the dataset as interpretable concepts, which we know by construction. In this case of properly chosen concepts, we find that using the bottleneck does not decrease, if not improves, the performance. This finding is in line with the rest of the paper.
> >
> > ## W3
> > With ‘free head’ we refer to a component that is not included in the CKA loss, see Section 3.2 for more information. We have made an attempt to write down the procedure from our original response more clearly in Appendix J (only in the latest revision of the paper).
> >
> >
> > Finally, we highly appreciate your constructive comments, and truly intend to answer your points precisely. We kindly invite you to take a look at these new results in the paper, and please let us know if we have not addressed any of your points accurately.

---

### Official Review · Reviewer_J9av · 2024-11-05

**Soundness:** 2
**Presentation:** 3
**Contribution:** 2
**Rating:** 5
**Confidence:** 3

**Summary:**

In an effort to develop more interpretable time-series forecasting models, the authors have combined a transformer-based architecture (Autoformer) with a concept bottleneck approach. The concepts do not correspond to any a priori annotations bur rather are derived either from an autoregressive model or from sample timestamps.  The authors encourage the network to "reason" using these concepts by adding an additional term to the loss function that captures the similarity between the model's internal representations and the precomputed concepts. The balance between prediction error and representational alignment (in the cost function) with the concepts is regulated through a single hyperparameter. Furthermore, the alignment scores with the different concepts (as captured by CKA) seem to make intuitive sense for many of the datasets (electricity usage and time of day for example). Overall, this is an interesting approach to a timely problem in the field. That said, there are some open questions about the approach that need to be addressed.

**Strengths:**

- The goal of the paper, the presentation, and the implementation details are clear
- The approach does not require costly annotations for concepts and can (arguably) be applied to any time-series data

**Weaknesses:**

- Looking at the qualitative results in Figure 9 and the summary in Table 1, this approach seems to do well for data that has a strong cyclical component (traffic and electricity). In fact, for all other datasets, the simpler AR model works best. How do you explain this? It seems like you get performance AND interpretability using an AR model, then why do you need a model with many more parameters? Maybe there are other datasets that could highlight the benefit of this approach (vs a simple AR based model) a little better? Perhaps I misunderstood something.
- It's hard to get an understanding of how the model is leveraging the concepts, especially since your results on hyper-parameter sensitivity (Table 5, Appendix F) are not the most intuitive; the first and second best settings of the alpha parameter are far apart (0.7 and 0.0). For pedagogical reasons, it might help to train the model on a synthetic dataset, constructed with the concepts (+noise) of your choice. Using a synthetic dataset might give the reader some more mechanistic intuition.

**Questions:**

- Is the optimal setting for the hyper-parameter dataset specific?

---

> ### Author Response · Authors · 2024-11-19
>
> Thanks for the useful comments! We hope that we can address the two worries you mention, and convince you to increase the scores slightly.
>
> (1) Regarding your point about the simpler AR model often being the best: yes, we agree this is fairly interesting. In fact, there has been much discussion about the benefits of Transformers for time series forecasting, because simpler models seem to outperform them for some datasets. While many works (e.g., [1] next to milestone approaches such as Informer, FedFormer, and others)  are in favour of employing Transformers for time series, others are not (e.g., [2], and the response from Huggingface [5]). Moreover, this is only a part of more general ongoing discussions regarding machine learning vs. statistical methods (see the influential Makridakis et al. [3], and often-mentioned Nixtla experiments [4]) which has already been a part of time-series forecasting literature for last couple of years, and likely will continue to be so.
>
> In our paper, we tried to sidestep all these discussions, and rather focus on the Transformer's interpretability. Therefore, we do not make claims about advantages of Transformers in all time series data, but do argue that IF they are used, then interpretability is a major issue that needs to be addressed.
>
> We focused our analyses on cases where the Transformer does outperform other models (e.g. the traffic and electricity dataset), and showed that our concept bottleneck framework is applicable. Therefore, we do not consider it a weakness that AR sometimes outperforms the Autoformer or not. We do test the framework for all these datasets, because they are often used in the time series literature (including the original Autoformer paper).
>
> (2) Regarding the optimal setting for the alpha hyper-parameter: we find that the model performance is not heavily dependent on alpha. Recall that alpha indicates the weight of the CKA term in the loss function, and we find that so long as the CKA term is not equal to the loss function (i.e. alpha = 1), then there is a term which pushes the model to learn to forecast well. This is in line with the overall results, including the bottleneck (alpha > 0) does not decrease the original model performance (alpha = 0), and this holds for all datasets. Additionally, we would like to point out that almost all results for alpha < 1 in Table 5 from Appendix F are within the same range by standard deviation, so the best and second-best settings do not carry that much of weight.
>
> [1] Niu, P., Zhou, T., Wang, X., Sun, L., & Jin, R. (2024). Attention as Robust Representation for Time Series Forecasting. ArXiv, abs/2402.05370.
>
> [2] Zeng, A., Chen, M., Zhang, L., & Xu, Q. (2022). Are Transformers Effective for Time Series Forecasting? AAAI Conference on Artificial Intelligence.
>
> [3] Makridakis, Spyros & Spiliotis, Evangelos & Assimakopoulos, Vassilis & Semenoglou, Artemios-Anargyros & Mulder, Gary & Nikolopoulos, Konstantinos. (2022). Statistical, machine learning and deep learning forecasting methods: Comparisons and ways forward. Journal of the Operational Research Society. 1-20. 10.1080/01605682.2022.2118629.
>
> [4] https://github.com/Nixtla/statsforecast/tree/main/experiments/m3
>
> [5] https://huggingface.co/blog/autoformer

---

> ### Author Response · Authors · 2024-11-27
> **Synthetic dataset**
>
> We have now performed the extra experiments you suggested with a synthetic dataset. To briefly recap: you proposed to train the model on a synthetic dataset, constructed with the concepts (plus noise) of our choice, to understand how the model leverages the concepts. In our earlier experiments, this was hard to achieve, because the best and second-best values of the hyperparameter alpha were not close in value, and therefore not intuitive (almost all results for $\alpha$ < 1 in Table 5 from Appendix F are within the same range by standard deviation, so the best and second-best settings do not carry that much weight. We included a new figure 10 in Appendix F that clearly illustrates this).
>
> The results from the new experiments are presented in Appendix I. We generate a time series dataset as the sum of different sine functions, and then train an Autoformer model with a bottleneck on the attention heads of the second layer. We vary the value of hyperparameter $\alpha$, and define each concept in the bottleneck as one of the underlying functions (for which we have the ground truth by construction).
>
> As expected, we find that the similarity between the bottleneck components and the concepts increases with increasing $\alpha$ (this is visible as the emergence of a yellow diagonal in layer 2 in Figure 23). At $\alpha=0$, there is no concept bottleneck and the similarity to the predefined concepts is minimal. At $\alpha=1.0$, the model is only optimized for similarity to the concepts, and the prediction performance is terrible. Interestingly, at $\alpha=0.8$, we hit a sweet spot where similarity to the predefined concepts is high and the prediction performance is also at its maximum.
>
> We believe these additional experiments help in understanding how the model leverages interpretable concepts. We would like to thank you again for the suggestion, and are curious to hear whether it is indeed exactly what you had in mind.

---

### Author Response · Authors · 2024-11-22
**Update paper: additional experiments on Vanilla Transformer**

To address the common weakness mentioned by multiple reviewers (regarding application to only one Transformer architecture), we included a new Appendix to the paper where we apply the framework to a different Transformer architecture: the vanilla Transformer. The Appendix is included at the end of the document, and the new parts are written in blue.

These additional results confirm the conclusions from the Autoformer experiments, in particular that the framework can be applied to a time series Transformer without having any significant impact on the overall model performance, while providing improved interpretability. Similar to the Autoformer model, the vanilla Transformer performs better than the AR model for the ‘Electricity’ and ‘Traffic’ dataset.

Since all time series Transformer architectures are derived from the vanilla Transformer, this deeper exploration with the framework highlights the general applicability of the framework. We kindly invite you to look at these additional results.

---

### Author Response · Authors · 2024-11-27
**Additional experiment: synthetic dataset**

We would like to notify the reviewers that we have included new results on a synthetic dataset in Appendix I, following the suggestion by Reviewer J9av. We believe these additional experiments help in understanding how the model leverages interpretable concepts. In summary, by increasing the weight of the CKA loss, we show that the bottleneck components become increasingly more similar to the (ground-truth) interpretable concepts, which helps with the forecasting task. The new parts are written in blue.

---

### Meta-Review · Area_Chair_d5MY · 2024-12-17

**Metareview:**

The paper proposes a novel framework for interpretability in time-series forecasting by integrating the concept bottleneck approach with the transformer-based Autoformer architecture. Instead of relying on predefined annotations, the model derives interpretable concepts from surrogate autoregressive (AR) models or sample timestamps. A Centered Kernel Alignment (CKA)-based loss term encourages the model’s internal representations to align with these concepts. The authors evaluate their approach on six benchmark datasets and demonstrate that the method maintains comparable predictive performance while improving transparency and enabling targeted interventions, such as handling temporal shifts.

Strengths

+ The application of the concept bottleneck model (CBM) to time-series forecasting is novel and well-motivated.
+ The paper introduces a creative use of CKA to align model representations with interpretable concepts.
+ The proposed framework integrates seamlessly with Autoformer without requiring costly annotations.
+ Intervention capabilities, such as handling temporal shifts, are demonstrated.
+ The approach maintains interpretability with minimal performance trade-offs.
+ The methodology is clear, and the paper is well-written and easy to follow.
+ The method is domain-agnostic, showing potential for broader applicability to other time-series tasks.

Weaknesses

+ The AR model outperforms the Autoformer with bottlenecks in four out of six datasets, questioning the need for a more complex model.
+ There is no comparison with other interpretability methods (e.g., SHAP, LIME, or attention-based visualizations).
+ The selection of interpretable concepts is heuristic, and the quality of these concepts is not thoroughly analyzed.
+ The analysis of hyperparameter sensitivity, such as the α weight, is limited and produces inconsistent results.
+ The CKA-based alignment encourages global similarity but does not capture fine-grained or localized temporal patterns.
+ The datasets used are relatively simple, primarily reflecting cyclical behaviors, limiting the generalizability of results.
+ There is no systematic study of the computational overhead, bottleneck locations, or scalability of the approach.
+ Visualizations and CKA analysis could be presented more clearly to enhance interpretability.

Some concerns have been addressed by the authors during the rebuttal period.

**Additional Comments On Reviewer Discussion:**

This is a borderline paper that receives two 5’s and two 6’s. One of the negative reviewers asked follow-up questions after the author response was posted. After discussion the reviewer was not convinced that the concerns on datasets, etc.

---

### Decision · Program_Chairs · 2025-01-22

Reject